# Inactivation of the Hippo tumor suppressor pathway promotes melanoma

Marc A. Vittoria[1], Nathan Kingston[2], Kristyna Kotynkova[1], Eric Xia [1], Rui Hong[3], Lee Huang[4], Shayna McDonald[1], Andrew Tilston-Lunel [2], Revati Darp[5], Joshua D. Campbell [3], Deborah Lang[4], Xiaowei Xu[6], Craig J. Ceol [5], Xaralabos Varelas [2] & Neil J. Ganem [1,3]✉

Melanoma is commonly driven by activating mutations in the MAP kinase BRAF; however, oncogenic BRAF alone is insufficient to promote melanomagenesis. Instead, its expression induces a transient proliferative burst that ultimately ceases with the development of benign nevi comprised of growth-arrested melanocytes. The tumor suppressive mechanisms that restrain nevus melanocyte proliferation remain poorly understood. Here we utilize cell and murine models to demonstrate that oncogenic BRAF leads to activation of the Hippo tumor suppressor pathway, both in melanocytes in vitro and nevus melanocytes in vivo. Mechanistically, we show that oncogenic BRAF promotes both ERK-dependent alterations in the actin cytoskeleton and whole-genome doubling events, which independently reduce RhoA activity to promote Hippo activation. We also demonstrate that functional impairment of the Hippo pathway enables oncogenic *BRAF*-expressing melanocytes to bypass nevus formation and rapidly form melanomas. Our data reveal that the Hippo pathway enforces the stable arrest of nevus melanocytes and represents a critical barrier to melanoma development.

[1] Department of Pharmacology & Experimental Therapeutics, Boston University School of Medicine, Boston, MA 02118, USA. [2] Department of Biochemistry, Boston University School of Medicine, Boston, MA 02118, USA. [3] Department of Medicine, Boston University School of Medicine, Boston, MA 02118, USA. [4] Department of Dermatology, Boston University School of Medicine, Boston, MA 02118, USA. [5] Program in Molecular Medicine, University of Massachusetts Medical School, Worcester, MA 01605, USA. [6] Department of Pathology and Laboratory Medicine, University of Pennsylvania Perelman School of Medicine, Philadelphia, PA 19104, USA. ✉email: nganem@bu.edu

Cutaneous melanoma arises from the malignant transformation of melanocytes, which are neural crest-derived cells mainly localized to the basal layer of the epidermis. When locally resected, melanoma is highly curable; however, melanoma is the most aggressive of all skin cancers and distant-stage disease is associated with significant mortality[1]. Unraveling the molecular features underlying the pathogenesis of cutaneous melanoma is essential for the development of preventative and therapeutic treatment strategies.

The vast majority of melanocytic neoplasms are initiated by oncogenic mutations in the mitogen-activated protein kinase (MAPK) pathway, with activating mutations in *BRAF* and *NRAS* occurring in ~50% and ~20% of cutaneous melanomas, respectively[2]. Within *BRAF*-mutant melanomas, the most common activating mutation results from a single amino acid substitution from a valine to a glutamic acid generating the constitutively active mutant $BRAF^{V600E}$[3,4]. Despite strongly inducing proliferative signaling, melanocyte-specific expression of $BRAF^{V600E}$ is insufficient to induce melanoma in multiple animal models; instead, $BRAF^{V600E}$ expression leads to the development of benign nevi (moles) comprised of growth-arrested melanocytes[5–10]. This is corroborated by clinical evidence as melanocytes within benign human nevi also frequently contain $BRAF^{V600E}$ mutations[11,12] and these melanocytic nevi rarely transform into melanoma (annual rate <0.0005%)[13]. Similarly, mutations within *NRAS* are commonly detected in congenital nevi and oncogenic *NRAS* expression in melanocytes in vivo does not rapidly yield melanoma[14–16]. Although the risk of any single melanocytic nevus transforming into melanoma is minimal, understanding how such transformations occur is paramount as roughly one-third of all melanomas co-exist with or arise from nevi[17].

These observations indicate that tumor suppression mechanisms restrain melanoma development following the acquisition of activating MAPK pathway mutations in melanocytes. A longstanding view is that strong oncogenic signals driven by mutations in MAPK pathway components lead to oncogene-induced cellular senescence (OIS), which safeguards against tumorigenesis[18–20]. Supporting this view, it has been demonstrated that expression of $BRAF^{V600E}$ in primary melanocytes in vitro induces an immediate cell cycle arrest and that these arrested melanocytes exhibit all of the hallmarks of oncogene-induced senescence: they become large, flat, vacuolar, express p16$^{INK4A}$, display senescence-associated β-galactosidase (SA-β-gal) activity, and have increased heterochromatic foci and DNA damage[18,21].

However, while it is clear that oncogene-induced senescence occurs in vitro, the extent to which this mechanism operates to ward off tumorigenesis in vivo remains unclear[22–24]. Several pieces of evidence argue against OIS as being the predominant mechanism restraining the proliferation of melanocytes harboring oncogenic mutations in vivo (extensively reviewed in ref. [16]). Most notably, oncogene expression (e.g., $BRAF^{V600E}$) in melanocytes does not induce an immediate proliferative block in vivo. Rather, these oncogenes initially induce proliferation, as evidenced by the clonal outgrowth of melanocytes that ultimately form a nevus, which requires many rounds of cell division. Furthermore, melanocytes lacking proteins known to enforce senescence, such as p16 and p53, retain the capacity to enter a growth-arrested state, as melanocytes in $Braf^{V600E}/Cdkn2a^{-/-}$ and $Braf^{V600E}/Trp53^{-/-}$ mouse models still primarily form nevi, with only a rare few melanocytes stochastically transforming into melanoma[7].

Collectively, these data suggest that additional tumor suppressive mechanisms have the capacity to restrain the proliferation of $Braf^{V600E}$-positive mouse melanocytes, independent of

inducing senescence. Recent modeling studies have led to the postulation that the growth arrest of nevus melanocytes is not solely due to oncogene activation and OIS in individual cells, but rather due to cells sensing and responding to their collective overgrowth, similar to what occurs in normal tissues[25]. This cell growth arrest is reminiscent of the arrest induced by activation of the Hippo tumor suppressor pathway, which is an evolutionarily conserved pathway known to regulate organ size. When the Hippo pathway is activated, the Hippo kinases LATS1/2 phosphorylate the transcriptional co-activators YAP (*YAP1*) and TAZ (*WWTR1*), resulting in their inactivation by nuclear exclusion and subsequent degradation[26,27]. In contrast, when the Hippo pathway is inactivated, YAP and TAZ are active and form DNA-binding complexes with the TEAD family of transcription factors, which act synergistically with AP-1 complexes to stimulate the expression of genes mediating entry into the S-phase and cell proliferation[28,29].

It is not known if Hippo pathway activation contributes to the growth arrest of nevus melanocytes. Moreover, while Hippo pathway inactivation has been suggested to promote cutaneous melanoma growth and invasion[30–32], it remains unknown whether Hippo inactivation is sufficient to induce cutaneous melanoma initiation and/or progression. Here, we use a combination of in vitro and in vivo model systems to examine the role of the Hippo tumor-suppressor pathway in restraining melanoma development.

## Results

**$BRAF^{V600E}$ expression activates the Hippo tumor-suppressor pathway in vitro**. We sought to examine if the expression of $BRAF^{V600E}$ is sufficient to induce activation of the Hippo tumor-suppressor pathway in cultured melanocytes. Previous studies using primary melanocytes have demonstrated that exogenous expression of oncogenic $BRAF^{V600E}$ leads to an immediate p53-dependent growth arrest[9,33]. We, therefore, developed a system in which $BRAF^{V600E}$ expression could be induced without an immediate cell cycle arrest in an attempt to explore Hippo pathway activation over multiple cell cycles. To do so, we generated a doxycycline-inducible system to permit controlled $BRAF^{V600E}$ expression in non-transformed Simian Virus 40 (SV-40) immortalized melanocytes (Mel-ST cells)[34]. Expression of the SV-40 early region, which encodes the small and large T viral antigens, imparts immortality to primary melanocytes via multiple mechanisms including impairment of the p53/Rb pathways[34]. Induction of $BRAF^{V600E}$ expression in Mel-ST cells increased the phosphorylation levels of the downstream kinases ERK and RSK, indicating that the cell model successfully hyperactivates MAPK signaling upon the addition of doxycycline (dox) (Fig. 1a).

To determine if $BRAF^{V600E}$ activates the Hippo tumor-suppressor pathway in vitro, we induced $BRAF^{V600E}$ expression and examined the relative levels of active, phosphorylated LATS1/2 at the hydrophobic motif (T1079)[27]. We found a significant increase in LATS phosphorylation following expression of oncogenic $BRAF^{V600E}$ (Fig. 1b). We then assessed total YAP phosphorylation (p-YAP) via phos-tag gel electrophoresis. We observed that $BRAF^{V600E}$ induction promoted phosphorylation of YAP at multiple sites (Fig. 1c). Consequently, expression of $BRAF^{V600E}$ led to nuclear exclusion of YAP and a corresponding decrease in the expression of the YAP target genes *CYR61* and *AMOTL2* (Fig. 1d, e and Supplementary Fig. S1A). The observed effects on LATS and YAP activity were due to $BRAF^{V600E}$, as overexpression of wild-type *BRAF* had no effect on LATS or YAP phosphorylation (Supplementary Fig. S1B–E). We further confirmed these results in multiple cell lines, including

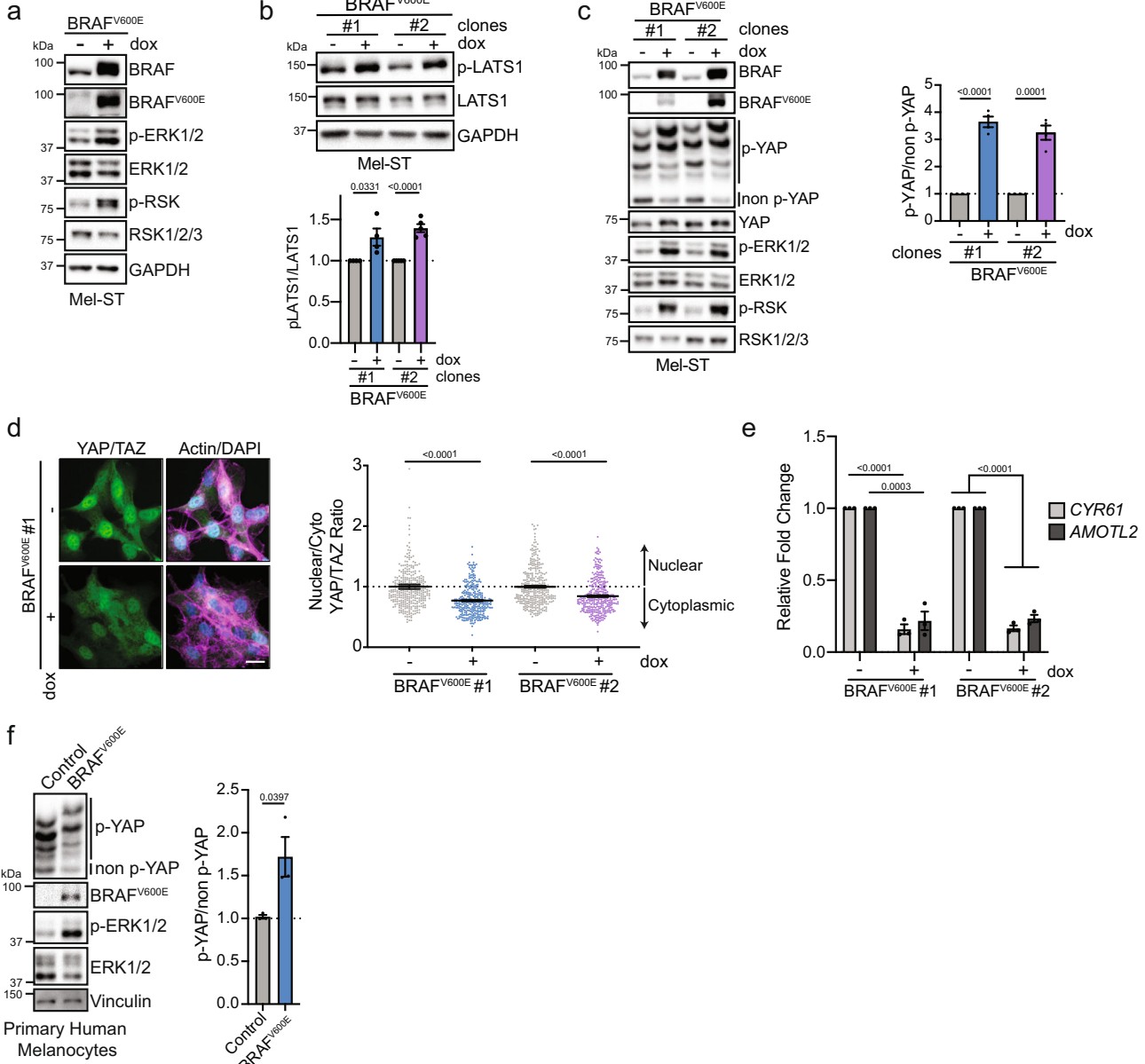

**Fig. 1 $BRAF^{V600E}$ activates the Hippo tumor-suppressor pathway. a** Representative immunoblot (IB) of dox-inducible $BRAF^{V600E}$ Mel-ST cells cultured ± dox for 24 h. **b** IB of two different dox-inducible $BRAF^{V600E}$ Mel-ST clones cultured ± dox for 24 h ($n \geq 4$ independent experiments, graph shows mean relative intensity ± SEM, two-tailed unpaired $t$ test). **c** Left, IB of two dox-inducible $BRAF^{V600E}$ Mel-ST clones cultured ± dox for 24 h; Right, intensity quantification of YAP phosphorylation from phos-tag gel ($n = 4$ independent experiments, graph shows mean relative intensity ± SEM, two-tailed unpaired $t$ test). **d** Left, representative immunofluorescence staining of YAP/TAZ (green) alone or merged with DNA (DAPI, blue) and actin (Phalloidin, magenta) in indicated $BRAF^{V600E}$ Mel-ST clone; right, quantification of nuclear to cytoplasmic ratio of mean YAP/TAZ fluorescence ($n > 300$ cells from three independent experiments, graph shows mean ± SEM, scale bar = 20 μm, two-tailed Mann–Whitney test). **e** Relative expression of indicated genes from RT-PCR in $BRAF^{V600E}$ Mel-ST clones cultured ± dox for 24 h ($n = 3$ independent experiments, graph shows mean ± SEM, two-tailed unpaired $t$ test). **f** Left, IB of primary human melanocytes infected with lentivirus that express control vector (H2B-GFP) or BRAF$^{V600E}$; right, intensity quantification of YAP phos-tag ($n = 3$ independent experiments, graph shows mean ± SEM, two-tailed unpaired $t$ test). Source Data are provided as a Source Data file.

non-immortalized primary adult human melanocytes with an intact p53 pathway (Fig. 1f and Supplementary Fig. S1F). Importantly, the observed effects of Hippo pathway activation were not limited to the expression of $BRAF^{V600E}$ alone, as we also found that inducible expression of oncogenic $NRAS^{Q61R}$ similarly activates the Hippo pathway (Supplementary Fig. S1G). Collectively, these data demonstrate that hyperstimulation of the MAPK signaling pathway through the expression of oncogenic $BRAF^{V600E}$ or $NRAS^{Q61R}$ leads to activation of the Hippo tumor-suppressor pathway in vitro.

**Growth-arrested melanocytes in benign nevi show evidence of Hippo pathway activation**. It is well established that melanocyte-specific expression of $Braf^{V600E}$ in animal models gives rise to benign nevi that harbor non-proliferating melanocytes[5,25]. We hypothesized that these $Braf^{V600E}$-positive melanocytes may also demonstrate evidence of Hippo pathway activation, similar to our in vitro results. To test this possibility, we analyzed a single-cell RNA sequencing dataset of whole-skin extracts collected at postnatal days 30 (P30) and 50 (P50) from wild-type control mice and tamoxifen-painted

$Tyr::CreER^{T2}/Braf^{CA}$ mice expressing active $Braf^{V600E}$[25]. We interrogated this dataset to examine if YAP/TAZ-dependent gene transcription was repressed in melanocytes expressing oncogenic $Braf^{V600E}$. Following dimensionality reduction and initial clustering, we identified the cluster representing melanocytes based on the expression of the melanocyte-lineage marker $Dct$ as previously reported (Fig. 2a)[25]. As expected, this cluster was found to have high expression of other melanocyte-lineage markers, notably $Mlana$ and $Mitf$ (Supplementary Fig. S2A). We then employed the variance-adjusted Mahalanobis (VAM) method, a scRNA-seq optimized approach to obtain accurate signaling pathway scores, to examine if YAP/TAZ-mediated gene expression was decreased in $Braf^{V600E}$-expressing mouse melanocytes relative to wild-type melanocytes utilizing previously published YAP/TAZ gene expression profiles[35–37]. The most basic analysis, where all single cells were binned by genotype, demonstrated that YAP/TAZ gene expression targets were significantly reduced in $Braf^{V600E}$-positive melanocytes compared to wild-type (Fig. 2b).

We then performed unsupervised clustering, which generated five melanocyte subdivisions, to clarify which unique populations of oncogenic $Braf^{V600E}$-expressing melanocytes exhibited the least YAP/TAZ activity (Fig. 2c). We theorized that clusters containing nevus melanocytes would be exclusively populated by cells isolated from $Braf^{V600E}$ mice, and be the primary melanocytic subtype isolated from 50-day-old mice. Based on these criteria, we identified clusters 0 and 1 as $Braf^{V600E}$-expressing melanocytes isolated from nevi (Fig. 2c and Supplementary Fig. S2B, S2C). In support of this prediction, expression of $Cdkn2a$, which is upregulated in nevus melanocytes, was found to be the highest in clusters 0 and 1, although $Cdkn2a$ read-depth was limited throughout all clusters (Supplementary Fig. S2D). Compared to all other melanocytes, nevus melanocytes (clusters 0 and 1) exhibited the lowest YAP/TAZ activity scores of any cluster, demonstrating that YAP/TAZ-mediated gene expression is reduced following oncogenic $Braf$ expression in mouse nevus melanocytes (Fig. 2d–f). Importantly, expression of Hippo pathway components remained unchanged regardless of genotype or cluster, suggesting decreased YAP/TAZ signaling was due to Hippo pathway activation, not altered expression of YAP/TAZ regulators (Supplementary Fig. S2E, F).

Not all melanocytes captured from $Braf^{V600E}$ mice exhibited low YAP/TAZ activity scores. Clusters 2 and 4, which contain an appreciable portion of melanocytes from both $Braf^{+/+}$ and $Braf^{V600E}$ mice, demonstrated much higher YAP/TAZ activity relative to nevus melanocytes (Supplementary Fig. S2G). However, within these clusters, YAP/TAZ activity was still decreased in $Braf^{V600E}$ melanocytes compared to wild-type cells. This suggests that cell-intrinsic mechanisms following $Braf^{V600E}$ expression are only partially leading to decreased YAP/TAZ activity and that other mechanisms, possibly cell-extrinsic cues, may play additional roles in vivo. We suspect cluster 3, the only cluster that did not exhibit this trend, may be comprised of proliferating, follicular melanocytes as this cluster mainly contains melanocytes isolated at P30 when most murine hair follicles are in anagen[25]. Taken together, these data reveal $Braf^{V600E}$-expressing mouse melanocytes largely exhibit decreased YAP/TAZ activity, with the most significant decreases found within nevus melanocytes, strongly implying the Hippo pathway becomes activated in response to $Braf^{V600E}$ expression and nevus formation in vivo. In support of these conclusions, immunofluorescence staining of three human benign nevi revealed YAP localization to be predominantly cytoplasmic and thus presumably inactivated in human nevus melanocytes, consistent with a previous study using a validated YAP antibody (Fig. 2g)[38].

## BRAF$^{V600E}$-induced Hippo activation restrains oncogenic melanocyte proliferation

We next investigated whether Hippo tumor-suppressor activation following $BRAF^{V600E}$ expression leads to reduced melanocyte proliferation in vitro. Population doubling assays demonstrated that expression of $BRAF^{V600E}$ reduced Mel-ST cell number ~30–40% relative to uninduced controls over a 4-day period, despite the fact these melanocytes were SV-40 immortalized (Fig. 3a). Live-cell imaging and proliferation assays revealed this was predominantly due to a proliferative arrest, rather than increased cell death (Supplementary Figs. S3A, B and 4A). To test whether the observed Hippo pathway activation induced by $BRAF^{V600E}$ expression was responsible for this proliferative defect, we used RNAi to knock down the LATS1/2 kinases in the context of $BRAF^{V600E}$ expression. We found that loss of LATS1/2 prevented YAP phosphorylation following induction of $BRAF^{V600E}$ and fully rescued cell growth and viability (Fig. 3b, c). We further found that inhibition of LATS1/2 with a potent small-molecule inhibitor also rescued cell growth (Fig. 3d)[39]. We then validated these findings using soft-agar growth assays. While parental Mel-ST cells and Mel-ST cells expressing $BRAF^{V600E}$ failed to efficiently grow under anchorage-independent conditions, Mel-ST cells expressing $BRAF^{V600E}$ together with a constitutively active YAP mutant (YAP-5SA) demonstrated significant colony formation (Fig. 3e and data not shown). These data reveal that functional inactivation of the Hippo pathway, through either LATS1/2 depletion/inhibition, or constitutive YAP activation, is sufficient to restore proliferation to $BRAF^{V600E}$-expressing immortalized melanocytes in vitro.

We sought to determine to what extent human melanoma cells inactivate the Hippo pathway. We first interrogated the The Cancer Genome Atlas (TCGA), where we found that co-heterozygous loss of $LATS1/2$ is observed in ~15% of human melanomas (Fig. 3f and Supplementary Fig. S3C, D). We then stained a panel of human melanoma samples for YAP localization utilizing SOX10 as a marker for melanoma cells and, in agreement with previous studies, found that multiple melanoma tumors exhibited strong nuclear YAP localization, suggesting Hippo pathway inactivation (Fig. 3g and Supplementary Fig. S3E)[31,38]. Collectively, these data reveal that a subset of oncogenic melanocytes during melanomagenesis will overcome or bypass Hippo pathway activation to regain proliferative capacity.

## BRAF$^{V600E}$-induced Hippo activation is ERK-dependent and partially mediated by changes in the actin cytoskeleton

We sought to understand the mechanisms through which $BRAF^{V600E}$ directly or indirectly activates the Hippo tumor-suppressor pathway. Oncogenic MAPK signaling has previously been shown to impair mitosis, and complete mitotic failure can lead to the generation of tetraploid cells that activate the Hippo pathway[40–43]. We speculated that the expression of $BRAF^{V600E}$ may lead to Hippo pathway activation by disrupting the normal completion of mitosis. To test this possibility, we performed live-cell imaging of doxycycline-inducible $BRAF^{V600E}$ Mel-ST cells stably expressing the chromosome marker histone 2B-GFP (H2B-GFP). We observed that upon entering mitosis, cells expressing $BRAF^{V600E}$ often exhibited widely oscillating chromosomes and were unable to maintain a tightly aligned metaphase plate relative to uninduced controls (Fig. 4a). These chromosome alignment defects impaired the ability of many cells to satisfy the spindle assembly checkpoint, and consequently a portion of the $BRAF^{V600E}$-expressing cells endured a significantly prolonged mitosis (Fig. 4a, b and Supplementary Fig. S4A, B). Cells that cannot satisfy the spindle assembly checkpoint either undergo mitotic cell death, or exit from mitosis without undergoing cell

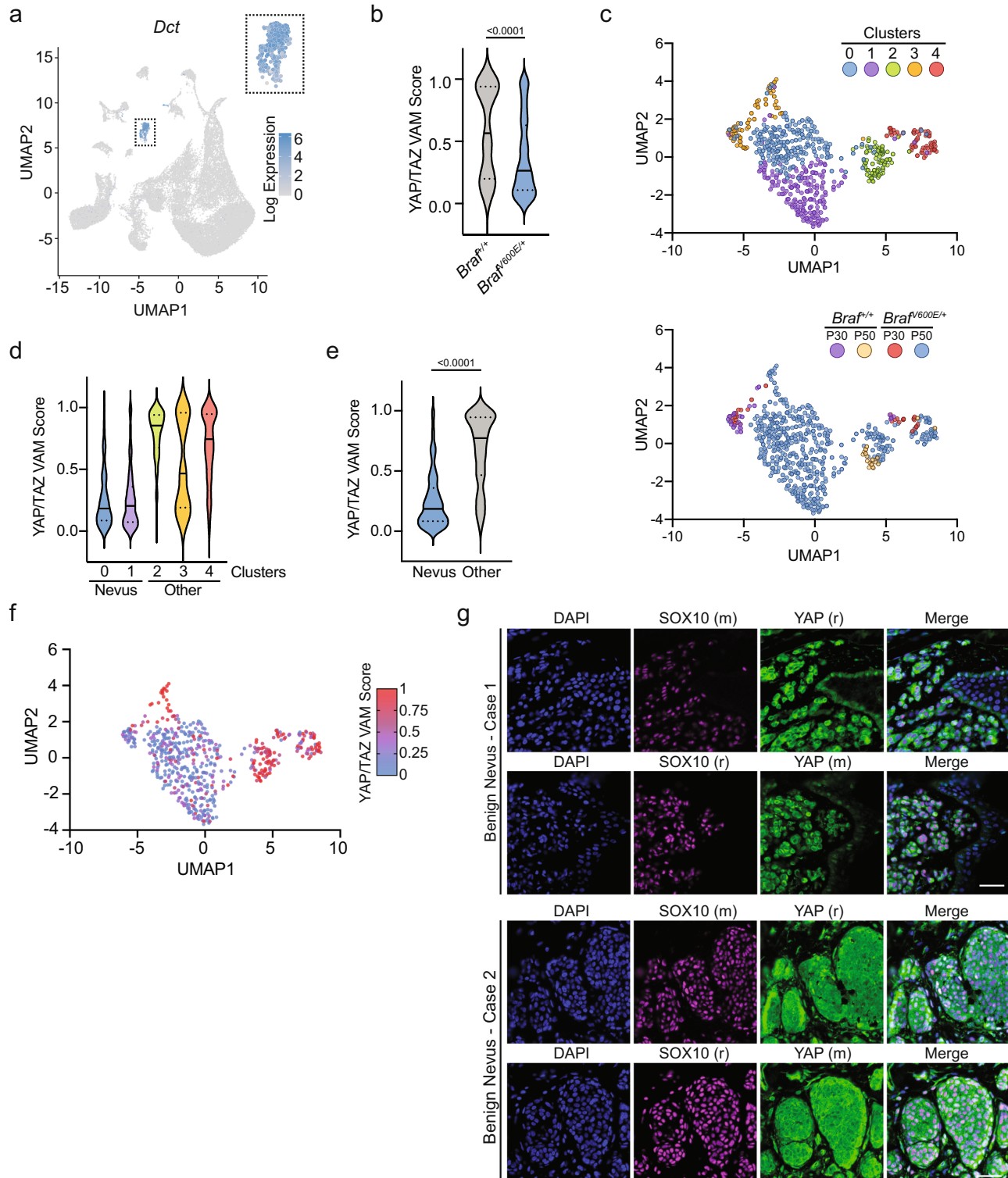

**Fig. 2 *Braf^{V600E}*-expressing nevus melanocytes display decreased YAP/TAZ signaling. a** UMAP of relative *Dct* expression of all single cells from nevus containing murine skin identifying a cluster of cells representing melanocytes from GSE154679. **b** YAP/TAZ VAM scores for melanocytes from indicated genotypes ($n = 46$ for *Braf^{+/+}*, $n = 543$ for *Braf^{V600E/+}*, two-tailed Mann–Whitney test). **c** Top, UMAP of melanocytes colored by subcluster; Bottom, UMAP of melanocytes colored by genotype and animal age ($n = 589$). **d** YAP/TAZ VAM score plotted by melanocyte subcluster. **e** YAP/TAZ VAM score comparing nevus (clusters 0, 1) and other melanocytes (clusters 2, 3, 4) (nevus $n = 408$, other $n = 181$, two-tailed Mann–Whitney test). **f** UMAP of melanocytes colored by gradient indicating YAP/TAZ VAM score. **g** Representative immunofluorescence staining of indicated proteins in two benign nevi cases with two different sets of antibodies, (r) = rabbit, (m) = mouse, DAPI (blue), YAP (green), SOX10 (magenta), scale bar = 50 μm. Source Data are provided as a Source Data file.

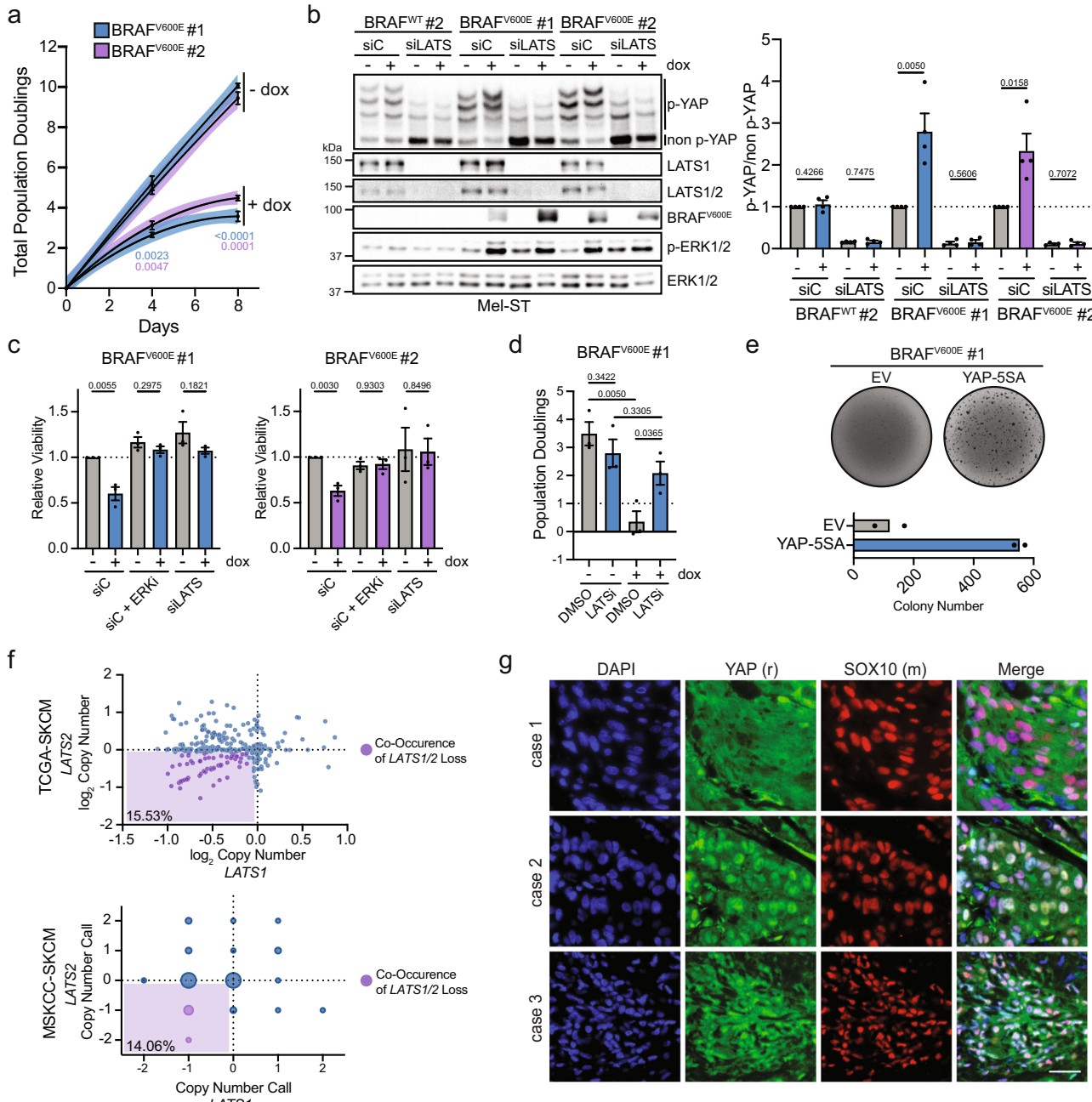

**Fig. 3 Oncogenic BRAF-induced Hippo pathway activation restrains melanocyte growth. a** Population doubling assay of two dox-inducible *BRAF^V600E* Mel-ST clones ± dox (*n* = 3 independent experiments, lines represent second-order polynomial non-linear fit, line color represents 95% confidence interval (dots show mean ± SEM, two-tailed unpaired *t* test). **b** Left, representative IB of two dox-inducible Mel-ST clones treated with control siRNA (siC) or LATS1 and LATS2 siRNAs (siLATS) ± dox for 24 h; right, intensity quantification of YAP phos-tag (*n* = 4 independent experiments, graph shows mean ± SEM, two-tailed unpaired *t* test). **c** Relative viability of Mel-ST cell lines treated with indicated siRNA or drugs for a 4-day period (*n* = 3 independent experiments in technical quintuplicate, graphs show mean ± SEM, two-tailed unpaired *t* test). **d** Number of population doublings of indicated dox-inducible Mel-ST clone over 4 days treated with either DMSO or a LATS1/2 inhibitor (LATSi), ± dox (*n* = 3 independent experiments, graph shows mean ± SEM, two-tailed unpaired *t* test). **e** Representative crystal violet stain of dox-inducible *BRAF^V600E* Mel-ST cell line expressing indicated genes grown in dox-containing soft agar with quantification below (*n* = 2 independent experiments in technical triplicate, graph shows mean). **f** Plot of log2 copy-number values from TCGA-SKCM or GISTIC 2.0 calls from MSKCC-SKCM databases for the genes *LATS1* and *LATS2*; bottom left percent is the frequency of *LATS1/2* co-heterozygous loss. **g** Representative immunofluorescence staining of indicated proteins in three human melanoma cases, scale bar = 25 µm). Source Data are provided as a Source Data file.

division, a phenomenon termed mitotic slippage[44]. Cells that undergo mitotic slippage often generate multinucleated tetraploid cells, and multinucleated melanocytes have been observed in human nevi[42,44,45]. We observed that upon induction of *BRAF^V600E* the number of mitoses producing tetraploid cells

increased significantly (control: 2.47%, induced: 24.75%) and was mainly driven by an increase in mitotic slippage (control: ~1%, induced: ~20%) (Fig. 4b and Supplementary Fig. S4B). These data demonstrate that *BRAF^V600E* can impair mitosis leading to mitotic slippage and the formation of multinucleated tetraploid

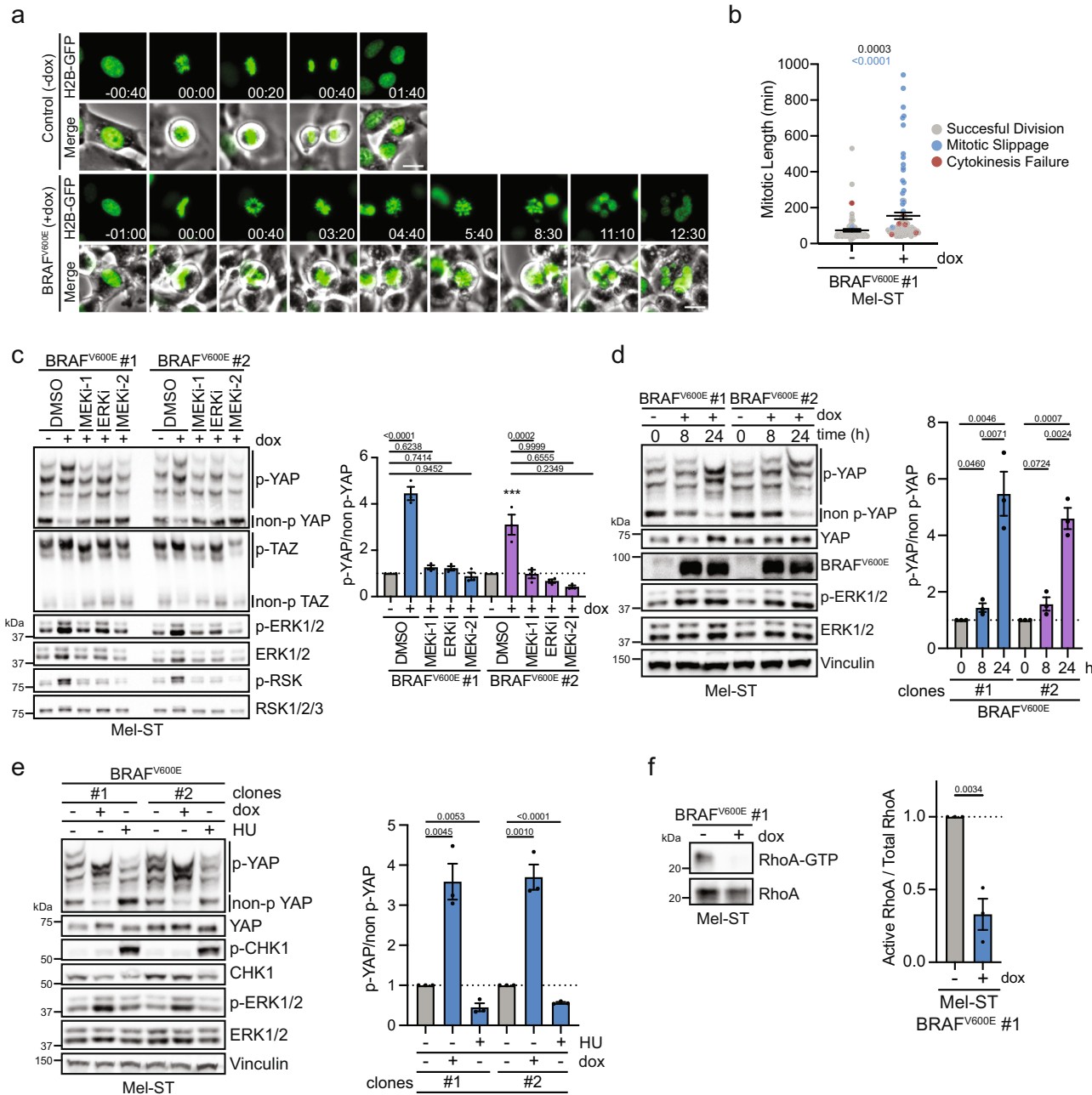

**Fig. 4 Prolonged MAPK activation leads to cytoskeletal defects and Hippo activation. a** Representative fluorescence and phase-contrast images from a live-cell video of dox-inducible *BRAF^V600E* Mel-ST cells expressing the chromosome marker H2B-GFP (green) cultured ± dox (scale bar = 25 μm, hh:mm). **b** Plot of mitotic duration and fate of individually tracked mitoses from (**a**) (*n* > 80 mitoses per condition from two independent experiments, graph shows mean ± SEM, dots represent individually tracked mitoses, black *P* value represent mitotic duration significance, two-tailed unpaired *t* test, blue *P* value represent significance for difference in frequency of whole-genome doubling events, two-sided Fisher's exact test). **c** Left, IB of indicated dox-inducible *BRAF^V600E* Mel-ST cell lines cultured ± dox for 24 h along with indicated drugs at the following doses: ERKi (20 nM), MEKi-1 (10 μM), MEKi-2 (20 nM); right, intensity quantification of YAP phos-tag (*n* = 3 independent experiments, graph shows mean ± SEM, one-way ANOVA with multiple comparisons). **d** Left, IB of dox-inducible *BRAF^V600E* Mel-ST cell lines cultured ± dox for indicated time; right, intensity quantification of YAP phos-tag (*n* = 3 independent experiments, graph shows mean ± SEM, two-tailed unpaired *t* test). **e** Left, representative IB of dox-inducible *BRAF^V600E* Mel-ST cell lines treated ± dox for 24 h or with 1 mM hydroxyurea for 6 h; right, intensity quantification of YAP phos-tag (*n* = 3 independent experiments, graph shows mean ± SEM, two-tailed unpaired *t* test). **f** Representative IB of RhoA-GTP pulldown in indicated dox-inducible *BRAF^V600E* Mel-ST cell line ± dox; right, intensity quantification of RhoA-GTP to total RhoA (*n* = 3 independent experiments, graph shows mean ± SEM, two-tailed unpaired *t* test). Source Data are provided as a Source Data file.

melanocytes in vitro. However, two lines of evidence suggested that mitotic errors leading to tetraploidization were not the major underlying driver of Hippo pathway activation in $BRAF^{V600E}$-expressing melanocytes. First, $BRAF^{V600E}$-expressing Mel-ST cells arrested in $G_1$ (via thymidine) or $G_2$ (via RO-3306-mediated CDK1 inhibition) still experienced Hippo activation despite their inability to become tetraploid (Supplementary Fig. S4C). Second, immunofluorescence experiments revealed that mononucleated diploid cells also exhibited decreased nuclear YAP/TAZ, demonstrating tetraploidization was not necessary to observe Hippo pathway activation (Fig. 1d).

We next investigated whether $BRAF^{V600E}$ specifically, or rather hyperactivation of the MAPK pathway generally, is responsible for Hippo pathway activation. We found that dampening of MAPK signaling via inhibition of the downstream kinases MEK1/2 or ERK1/2 fully prevented Hippo pathway activation, as measured by YAP/TAZ phosphorylation status, in $BRAF^{V600E}$-inducible cell lines (Fig. 4c and Supplementary Fig. S5A). These data demonstrated that Hippo pathway activation is entirely mediated by general hyperactivation of MAPK signaling and requires factors downstream of ERK. These data also discounted the possibility that oncogenic $BRAF^{V600E}$ activates Hippo signaling via direct phosphorylation of key Hippo pathway components.

We noted that phosphorylation of YAP following $BRAF^{V600E}$ expression required sustained MAPK stimulation over a period of at least 12–16 h, as transient MAPK activation only minimally affected YAP phosphorylation (Fig. 4d and Supplementary Fig. S5B). We speculated that mounting oncogene-induced replication stress may be promoting Hippo pathway activation; however, induction of replication stress by hydroxyurea treatment alone was not sufficient to activate the Hippo pathway (Fig. 4e). Alternatively, it has been demonstrated that oncogenic activation of the MAPK pathway dramatically alters actomyosin cytoskeletal contractility and reduces RhoA activity in an ERK1/2-dependent manner[46–48]. Reductions in active RhoA are well known to promote Hippo pathway activation and, furthermore, ERK1/2-dependent cytoskeletal changes have previously been shown to modulate YAP/TAZ activity in melanoma cell lines[40,49,50]. We, therefore, posited that the reduction of RhoA activity may represent a mechanism by which $BRAF^{V600E}$-expressing cells activate the Hippo pathway in vitro. Indeed, we observed that there was a significant reduction in the number of actin stress fibers in cells following activation of $BRAF^{V600E}$, indicating reduced RhoA activity (Supplementary Fig. S5C, D). We then directly measured RhoA activity via pull-down assay, which revealed significantly decreased levels of active RhoA in Mel-ST cells expressing $BRAF^{V600E}$ relative to controls (Fig. 4f). These data suggest that $BRAF^{V600E}$-induced Hippo pathway activation is at least partially mediated by prolonged MAPK hyperstimulation leading to ERK1/2-dependent cytoskeletal dysregulation. Supporting this view, endogenous $Braf^{V600E}$ expression in mouse embryonic fibroblasts has been shown to drastically reduce actin stress fibers, and a recent study has also demonstrated that expression of $BRAF^{V600E}$ in RPE-1 cells leads to decreased RhoA activity in vitro and cytokinesis failure in zebrafish (Darp et al. unpublished)[51].

**$Lats1/2^{-/-}$ deletion promotes melanomagenesis.** Our data suggested that functional inactivation of the Hippo tumor-suppressor pathway may enable $BRAF^{V600E}$-expressing melanocytes to evade growth arrest and facilitate melanoma development. To test this, we generated mice carrying floxed alleles of both Lats1 and Lats2[52] with $Tyr::CreER^{T2}$ to allow for inducible, melanocyte-specific inactivation of the Hippo pathway

($Tyr::CreER^{T2}/Lats1^{f/f}/Lats2^{f/f}$). Deletion of LATS1/2 is well established to completely abrogate the Hippo pathway, and co-heterozygous loss of LATS1/2 is observed in ~15% of human melanomas, making deletion of Lats1/2 clinically relevant (Fig. 3f and Supplementary Fig. S3C, D)[2,53]. We also crossed $Tyr::CreER^{T2}/Lats1^{f/f}/Lats2^{f/f}$ ($Lats1/2^{-/-}$) mice with mice expressing the Cre-activatable oncogenic Braf allele ($Braf^{CA/+}$), generating $Tyr::CreER^{T2}/Braf^{CA}/Lats1^{f/f}/Lats2^{f/f}$ ($Braf^{V600E}/Lats1/2^{-/-}$) mice (Fig. 5a and Supplementary Fig. S6A). We confirmed the melanocytic specificity of our $Tyr::CreER^{T2}$ expressing mice via incorporation of a fluorescent lineage trace ($YFP^{LSL}$), whose expression was only observed in cells that co-stained for melanocyte markers (Supplementary Fig. S6B).

We observed that $Braf^{V600E}/Lats1/2^{-/-}$ mice were highly prone to developing spontaneous dermal tumors within weeks after birth, even without topical 4-hydroxytamoxifen (4-HT) administration. A similar melanoma mouse model, $Tyr::CreER^{T2}/Braf^{CA}/Pten^{f/f}$ ($Braf^{V600E}/Pten^{-/-}$), has also been shown to be prone to spontaneous melanoma formation in the absence of topical 4-HT, due to leakiness of the inducible Cre recombinase[54,55]. This suggests that deletion of Lats1/2 plays a major role in promoting melanoma development, as $Braf^{V600E}$ expression alone in murine melanocytes does not generate tumors[5,25] (Supplementary Fig. S6A). In the few mice where spontaneous tumorigenesis was absent or delayed, 4-HT administration to $Braf^{V600E}/Lats1/2^{-/-}$ flanks resulted in the potent form of tumors which appeared histologically similar to the spontaneously arising neoplasms (Fig. 5b, c and Supplementary Fig. S6C). These tumors exhibited strong nuclear YAP/TAZ staining, indicating Hippo inactivation, and positively stained for the melanocytic markers SOX10 and S100 (Fig. 5d). SOX10 staining was nuclear and homogenous whereas S100 staining was weakly heterogeneous. Subsequent histopathologic analysis by a dermatopathologist confirmed these infiltrative, spindle cell tumors to be mouse melanoma. Unlike other $Braf^{V600E}$-driven mouse melanoma models (e.g., $Braf^{V600E}/Cdkn2a^{-/-}$, $Braf^{V600E}/Trp53^{-/-}$), which still mainly induce nevus formation, we were unable to appreciate any obvious nevogenesis in $Braf^{V600E}/Lats1/2^{-/-}$ mice. These data imply oncogenic $Braf^{V600E}$-positive melanocytes may be incapable of entering an enduring growth arrest without a functional Hippo tumor-suppressor pathway.

We also investigated the consequences of Lats1/2 loss in melanocytes in the absence of oncogenic Braf. We found that following melanocyte-specific deletion of Lats1/2, mice exhibited no obvious hyperpigmentation, yet still rapidly developed cutaneous tumors with 100% penetrance after 4–5 weeks (Fig. 5e–g and Supplementary Fig. S6C–E). Co-heterozygous deletion of Lats1/2 also promoted cutaneous tumorigenesis, albeit at prolonged time scales (Supplementary Fig. S6F). Analysis of $Lats1/2^{-/-}$ tumor sections revealed non-pigmented neoplasms which were remarkably similar to invasive $Braf^{V600E}/Lats1/2^{-/-}$ mouse tumors, exhibited a comparable staining profile, and were subsequently diagnosed as mouse melanoma (Fig. 5h). Previous studies have identified TEAD and AP-1 transcription factors as major regulators of the melanoma invasive state, which is marked by dedifferentiation and loss of pigmentation signatures[56,57]. We hypothesized that YAP/TAZ-TEAD activation, driven by Lats1/2 deletion, may enable melanocytes to directly access this invasive gene program explaining our observed lack of pigmentation. In support of this hypothesis, both $Braf^{V600E}/Lats1/2^{-/-}$ and $Lats1/2^{-/-}$ mouse melanomas exhibited markedly low staining for mature, differentiated melanocyte markers (Supplementary Fig. S6G).

Given $Lats1/2^{-/-}$ tumors did not exhibit overt signs of pigmentation, we sought to generate additional data to validate that these neoplasms were melanocytic in origin. It has recently been demonstrated that initiation of the hair follicle cycle, via depilation, strongly promotes melanocyte transformation in

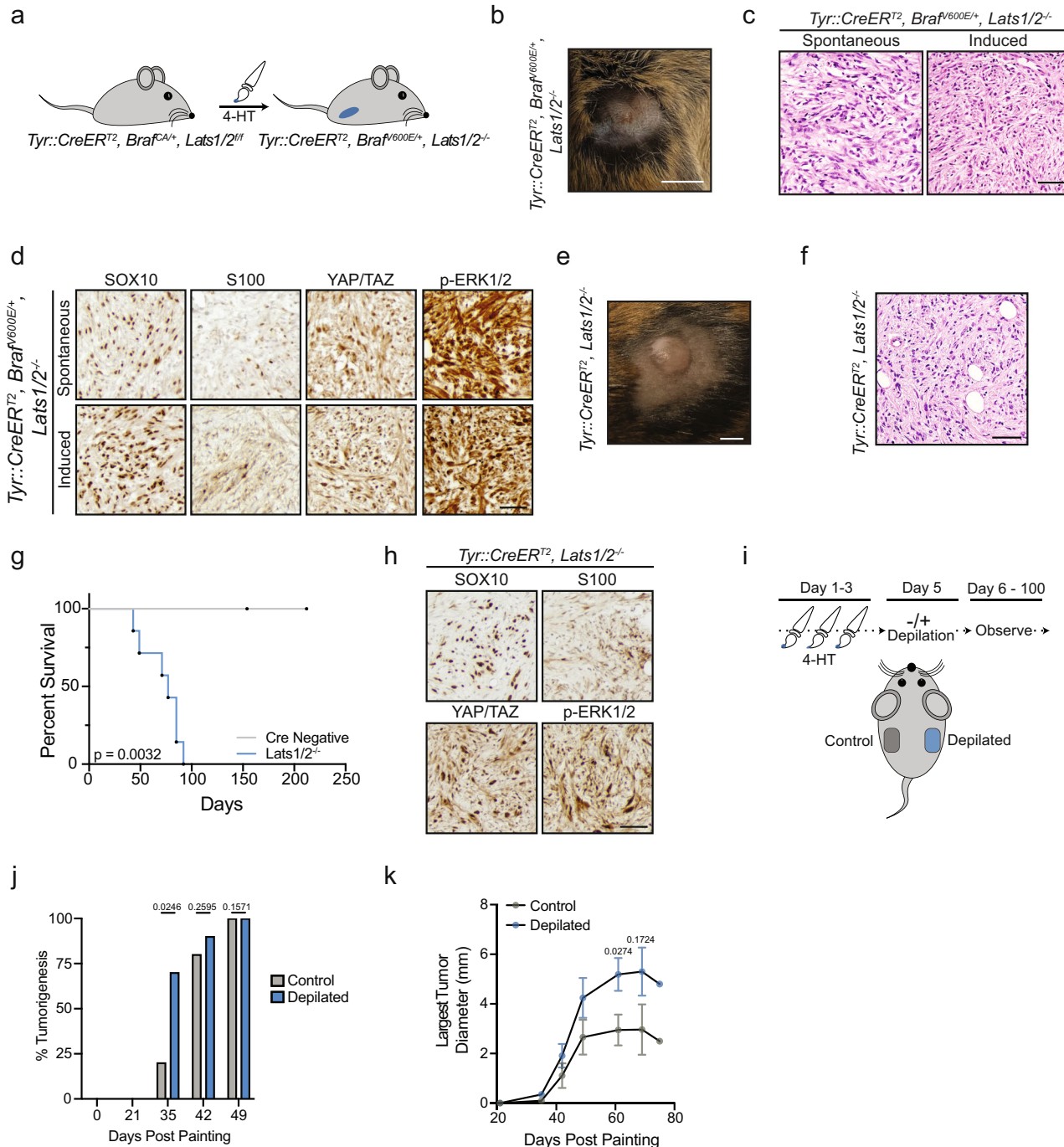

**Fig. 5 *Lats1/2* deletion promotes melanoma formation. a** Cartoon depicting 4-HT painting experiment. **b** Representative image of induced *Braf^V600E/Lats1/2^{-/-}* mouse tumor, scale bar = 0.5 cm. **c** Representative hematoxylin and eosin staining of spontaneous or induced tumors from *Braf^V600E/Lats1/2^{-/-}* mice, scale bar = 50 μm (*n* > 3 spontaneous tumors, *n* = 2 induced tumors from *Braf^V600E/Lats1/2^{-/-}* mice). **d** Representative IHC of indicated proteins, scale bar = 40 μm (*n* > 3 spontaneous tumors, *n* = 2 induced tumors from *Braf^V600E/Lats1/2^{-/-}* mice). **e** Representative image of induced *Lats1/2^{-/-}* mouse tumor, scale bar = 0.5 cm. **f** Representative hematoxylin and eosin staining of induced flank tumor from *Lats1/2^{-/-}* mouse, scale bar = 50 μm (*n* > 6 independent mice with similar results). **g** Survival plot from painting to study endpoint of *Lats1/2^{-/-}* mice with 4-HT painted on flanks (Cre negative *n* = 4, *Lats1/2^{-/-}* *n* = 7, log-rank test). **h** Representative IHC of indicated proteins, scale bar = 40 μm (*n* ≥ 3 *Lats1/2^{-/-}* mice). **i** Cartoon depicting depilation experiment in *Lats1/2^{-/-}* mice. **j** Percent palpable tumorigenesis in *Lats1/2^{-/-}* mice at indicated time points (*n* = 10 mice, two-tailed unpaired *t* test). **k** Diameter of largest tumor on each animal measured at indicated time points (*n* ≥ 3 tumors measured at each timepoint with error bars ± SEM, ten mice total, two-tailed unpaired t test). Source Data are provided as a Source Data file.

*Braf^V600E/Pten^{-/-}* mice[58,59]. We leveraged this unique characteristic of mouse melanomagenesis to test if *Lats1/2^{-/-}* tumor formation was also promoted by depilation, suggesting a melanocytic origin. We induced loss of *Lats1/2* on opposing mouse flanks and then depilated only one flank so as to compare the tumorigenic rate from depilated and non-depilated regions (Fig. 5i). We observed that skin regions depilated following 4-HT treatment demonstrated significantly faster tumorigenesis, with palpable tumors observed in 70% of depilated areas compared to 20% of non-depilated areas about one month following treatment

(Fig. 5j). Not only did tumors appear faster in depilated areas, but these tumors also grew significantly larger (Fig. 5k). Collectively, our data demonstrate that melanocyte-specific loss of *Lats1/2* alone, or in conjunction with oncogenic *Braf* expression, promotes mouse melanocyte transformation and the formation of mouse melanoma.

We then investigated whether Hippo pathway inactivation was also occurring in other mouse models of melanoma. We performed gene set enrichment analyses (GSEA) using gene expression data collected from benign and transformed melanocytes from $Braf^{V600E}/Cdkn2a^{-/-}$ and $Braf^{V600E}/Cdkn2a^{-/-}/Lkb1^{-/-}$ mice[7]. GSEA revealed that YAP/TAZ gene sets were significantly enriched in murine melanoma cells as compared to both arrested nevus melanocytes and proliferating non-tumorigenic melanocytes (Fig. 6a and Supplementary Fig. S7A–D). These data reveal that the Hippo tumor-suppressor pathway becomes attenuated as these murine melanocytes transform into melanoma.

**Active YAP drives melanoma development**. We aimed to further define how deletion of *Lats1/2* promotes melanoma development in vivo. While it is well-described that *Lats1/2* loss functionally inactivates the Hippo pathway and leads to the activation of YAP and TAZ, LATS1/2 can also impinge upon additional signaling pathways that promote tumor development. For example, recent research has revealed inactivation of the Hippo pathway can promote mTOR signaling via multiple routes including YAP-driven expression of a micro-RNA, miR-29, which targets PTEN mRNA for silencing[60–62]. However, we detected no observable changes in PTEN protein level following either RNAi-mediated knockdown of LATS1/2 or expression of constitutively active YAP (YAP-5SA) or TAZ (TAZ-4SA) in Mel-ST cells (Fig. 6b, c). We also could not appreciate any significant change in phosphorylated S6 levels following *LATS1/2* silencing (Fig. 6b). Furthermore, examination of $Lats1/2^{-/-}$ tumors revealed PTEN remained strongly expressed in vivo (Fig. 6d). These data reveal that loss of *Lats1/2* is not driving melanomagenesis by activating mTOR via miRNA-mediated depletion of PTEN. It has also been demonstrated that active LATS2 can bind and inhibit MDM2 leading to increased p53 protein levels[63], raising the possibility that deletion of *Lats1/2* leads to decreases in p53, which may facilitate $Braf^{V600E}$-driven murine melanomagenesis[64]. Discounting this, we found that p53 still accumulates in $Lats1/2^{-/-}$ tumors (Fig. 6d). Previous studies have also demonstrated that *Lats1/2* knockout can induce cytokinesis failure and whole-genome doubling (WGD) in MEFs. Since WGD is well known to facilitate tumorigenesis, and a significant fraction of human melanomas are WGD, we speculated that *Lats1/2* loss may drive tumorigenesis in vivo through an initial WGD event. To test this possibility, we performed copy-number analysis using ultra-low pass whole-genome sequencing (ULP-WGS) to examine if genomic alterations, such as WGD events, were enriched in $Braf^{V600E}/Lats1/2^{-/-}$ tumors relative to $Braf^{V600E}/Pten^{-/-}$ tumors. However, sequencing revealed that all tumors from these models were diploid, excluding the possibility that *Lats1/2* deletion was primarily inducing melanoma through a WGD intermediate (Supplementary Fig. S8A).

We then assessed whether YAP activation alone was sufficient to promote melanoma development. To do so, we generated a transgenic zebrafish model that expresses constitutively active YAP (YAP-5SA) in zebrafish melanocytes utilizing the mini-CoopR system[65]. These *Tg(mitfa:YAP-5SA)* zebrafish rapidly developed pigmented fish melanoma (Fig. 6e), demonstrating that constitutively active YAP is sufficient to induce melanoma development in a zebrafish model.

As activation of YAP/TAZ signaling was observed to promote melanomagenesis, we investigated whether depletion of YAP/

TAZ could inhibit melanoma cell growth. RNAi-mediated knockdown of YAP/TAZ in the $Braf^{V600E}/Pten^{-/-}$ mouse melanoma cell line (D4M.3A) resulted in significantly decreased viability as compared to immortalized melanocytes (Supplementary Fig. S7E). We further explored YAP/TAZ dependency in a panel of human melanoma cell lines utilizing the pan-TEAD inhibitor MGH-CP1 and found that a number of melanoma cell lines were strikingly sensitive to inhibition of YAP/TAZ activity (Fig. 6f and Supplementary Fig. S7F)[66]. While *YAP1* is not commonly mutated in human melanoma, *YAP1* amplifications and mutations have been observed, and YAP staining in primary melanoma has been shown to significantly correlate with reduced patient survival[31,38,67]. Together, these data implicate YAP as a cutaneous melanoma oncogene and novel therapeutic target to further explore in the treatment of human melanoma.

## Discussion

Discerning the molecular pathways that govern the growth arrest of nevus melanocytes, and how melanocytes ultimately overcome these barriers, is critical to fully understanding the mechanisms of melanomagenesis. A considerable body of work supports a role for OIS in preventing tumorigenic growth of melanocytes in vitro and in vivo; however, expanding lines of evidence demonstrate melanocytes within nevi retain proliferative capacity, in conflict with the absolute growth arrest implied by OIS[16]. Indeed, up to 30% of melanomas are predicted to arise from pre-cursor nevi[17]. These observations highlight nevi as being the product of "stable clonal expansion" rather than senescence[16]. Several genetic alterations that have the capacity to overcome the stable growth arrest of melanocytes expressing oncogenic MAPK mutations have been identified, including *CDKN2A, PTEN*, and *TP53*[3,5–7,64]. While deletion of *Cdkn2a* and *Trp53* with $Braf^{V600E}$ expression induces murine melanomagenesis, the vast majority of melanocytes still enter a stable growth arrest to form nevi[7]. Further, recent murine single-cell sequencing analyses reveal $Braf^{V600E}$-positive nevus melanocytes do not exhibit expression of senescence signatures[25]. Together, these discoveries suggest additional unidentified tumor suppressive mechanisms exist to enforce the arrest of nevus melanocytes.

We discovered that expression of $BRAF^{V600E}$ promotes activation of the Hippo tumor-suppressor pathway across multiple cell lines and that $Braf^{V600E}$-positive murine nevus melanocytes display significantly decreased YAP/TAZ signaling in vivo. Our data demonstrate that oncogenic *BRAF* expression induces Hippo pathway activation and cell cycle arrest in vitro by way of a cell-intrinsic mechanism, in which hyperactive MAPK signaling alters the cytoskeleton in part through decreased RhoA signaling and thus indirectly leads to Hippo pathway activation. However, our data also suggest that cell-extrinsic cues, perhaps secondary to the nevus microenvironment, may also be impinging upon YAP/TAZ signaling in vivo. Our single-cell sequencing analysis of mouse melanocytes indicated that while $Braf^{V600E}$-expression leads to a general repression of YAP/TAZ signaling in most melanocytes, the effect is strongest in nevus melanocytes (Fig. 2d). A recent modeling study has proposed that the growth arrest of nevus melanocytes may be due to cells sensing and responding to their collective overgrowth, rather than cell-autonomous mechanisms[25]. This growth arrest model is highly analogous to the collective cell processes mediating contact inhibition and organ growth, which are governed by the Hippo pathway. It is therefore tempting to speculate that melanocyte overgrowth induced by oncogene expression significantly increases melanocyte density over multiple rounds of division, ultimately promoting mounting activation of the Hippo tumor-suppressor pathway, eventual growth arrest, and nevus formation (Fig. 6g). If

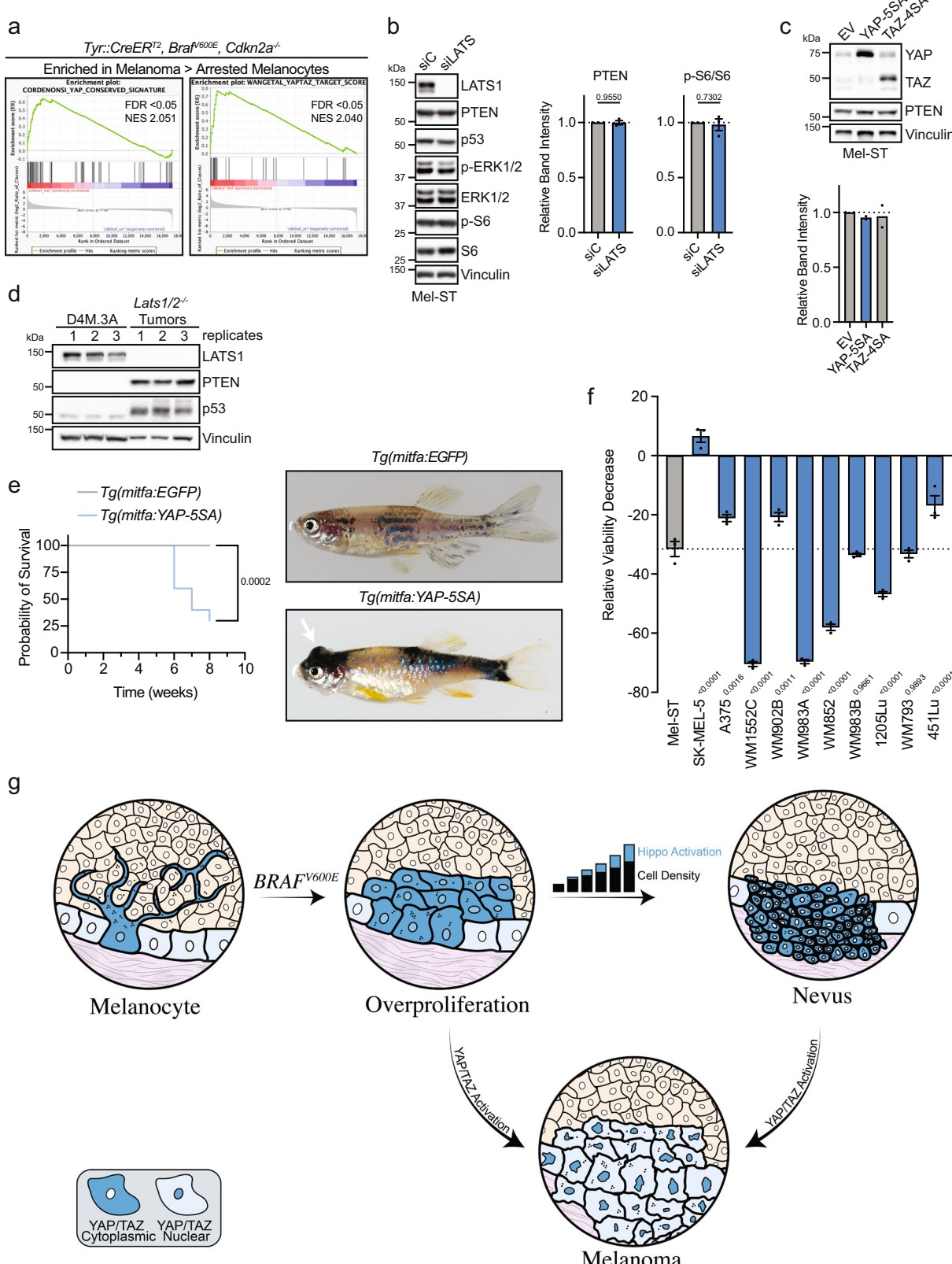

true, one would predict Hippo pathway activation would only engage after a nevus passes a critical size threshold and that nevus melanocyte growth should be restored by reducing local melanocyte density. Intriguingly, nevi that are partially resected have been observed to regain proliferative capacity and the majority of these recurrent nevi did not grow beyond the limits of the original surgical scar, suggesting these nevus melanocytes arrest once they attain a similar size[68,69]. While growth arrest of oncogene-expressing melanocytes is conspicuous due the formation of pigmented nevi, it is also possible that oncogene-induced growth arrest of cells from other tissues is similarly mediated by activation of the Hippo pathway.

**Fig. 6 YAP activation promotes melanomagenesis. a** GSEA performed on GSE61750 comparing enrichment of YAP/TAZ signatures in melanoma to arrested melanocytes in indicated genotype (see Supplementary Fig. S7A, S7D). **b** Left, representative IB of Mel-ST parental cell line treated with control siRNA (siC) or LATS1 and LATS2 siRNAs (siLATS); right, quantification of PTEN and p-S6 proteins compared to loading control or total protein ($n = 3$ independent experiments, graphs show mean ± SEM, two-tailed unpaired $t$ test). **c** Top, representative IB of Mel-ST parental cell line expressing indicated plasmids, EV: empty vector; bottom, quantification of PTEN compared to loading control ($n = 2$ independent experiments, graph shows mean). **d** IB comparing D4M.3 A cells and $Lats1/2^{-/-}$ tumor lysates where numbers represent replicates or different tumors, respectively ($n = 3$ independent experiments or mice). **e** Left, survival curve of zebrafish with indicated genotypes; right, representative images of zebrafish with indicated genotype ($n = 14$ independent EGFP fish, $n = 10$ independent YAP-5SA fish, log-rank test). **f** Graph shows relative viability decrease (%) in indicated cell lines treated with 10 μM MGH-CP1 for 4 days ($n = 3$ independent experiments in ≥technical quintuplate, graph shows mean ± SEM, one-way ANOVA with Dunnett's multiple comparisons test). **g** Schematic model in which we propose overproliferation of melanocytes following acquisition of oncogenic $BRAF^{V600E}$ leads to increases in local melanocyte density, which subsequently promotes Hippo pathway activation and contributes to growth arrest and nevus formation. Functional impairment of the Hippo pathway, leading to YAP/TAZ activation, enables melanocytes expressing $BRAF^{V600E}$ to evade stable growth arrest and promote the development of melanoma. Source Data are provided as a Source Data file.

We also demonstrated that functional impairment of the Hippo pathway in melanocytes in vivo, either through deletion of $Lats1/2$ in mice or expression of constitutively active $YAP1$ in zebrafish, promotes cutaneous melanomagenesis. This finding has clinical relevance as co-heterozygous loss of $LATS1/2$ and amplification of $YAP1$ are observed in primary and metastatic human melanoma[31,32,38]. However, it should be noted that functional impairment of the Hippo pathway alone is not observed in human melanoma, and thus our model only partially replicates features of human tumors. Moving forward, it will be important to define additional mechanisms by which human melanoma cells circumvent the Hippo pathway to activate YAP[38,70,71]. For example, >80% of uveal melanomas and ~6% of cutaneous melanomas have mutations in GNAQ/GNA1 proteins, which stimulate RhoA activity and activate YAP/TAZ independent of any alterations in $LATS1/2$[72–74].

Our live-cell imaging also revealed that induction of oncogenic $BRAF$ promotes chromosome alignment defects, prolonged mitosis, and mitotic or cytokinetic failures that lead to whole-genome doubling (WGD). Cells that have experienced a WGD (WGD$^+$ cells) are genomically unstable and tumorigenic, and their contribution to human cancer is significant[75–77]. Oncogenic $BRAF$ may therefore facilitate tumorigenesis not only by activating MAPK signaling, but also by increasing the baseline level of oncogenic WGD$^+$ cells. However, while ~30–40% of all human melanomas, and the vast majority of human metastatic melanomas, demonstrate evidence of WGD[78], our mouse melanoma models failed to exhibit this genomic feature (Supplementary Fig. S8). This is likely because the mouse models are already potently tumorigenic without the need for the additional oncogenic effects imparted by a WGD; nevertheless, the lack of any appreciable WGD in murine melanomas illustrates that these models have limitations in recapitulating the human disease[45,76–78].

It is believed that activating MAPK mutations are critical for human melanocyte transformation; however, we found that murine melanocytes lacking $Lats1/2$ rapidly developed into melanomas without the initial presence of any other genetic alterations that stimulate the MAPK pathway. Questions therefore remain as to how Hippo pathway inactivation alone can so strongly promote melanoma development in mice. We observed that melanomas generated from $Lats1/2^{-/-}$ mice exhibited strong p-ERK staining. This suggests that the $Lats1/2^{-/-}$ melanocytes likely evolved to hyperactivate the MAPK pathway during their transformation, either through the acquisition of additional genetic alterations or, potentially, by stimulating or emulating MAPK activity through YAP/TAZ-dependent transcriptomic changes[73]. Indeed, it has been observed that one common resistance mechanism to Vemurafenib, a $BRAF^{V600E}$-specific inhibitor, is the amplification and/or activation of YAP[49,70,79]. This

suggests that YAP activity can compensate for loss of $BRAF^{V600E}$ signaling. Given MAPK inhibitor resistance remains a significant component of treatment failure, our data suggests that co-targeting MAPK and YAP-TEAD signaling could simultaneously prevent resistance[70] and decrease melanoma cell viability[32,80].

In summary, we demonstrate that activation of the Hippo tumor-suppressor pathway promotes melanocyte growth arrest in response to the expression of oncogenic $BRAF^{V600E}$. Functional impairment of the Hippo pathway potently induces melanocyte growth in vitro and tumor development in vivo in multiple model organisms of melanomagenesis. Collectively, our data implicate the Hippo pathway as an important melanoma tumor suppressor and highlight YAP/TAZ as promising therapeutic targets to investigate for the treatment of human melanoma.

## Methods

**Cell culture**. Immortalized melanocyte (Mel-ST) cells were a gift from the lab of Dr. Robert Weinberg. Mel-ST cells, and all derivative cell lines generated in this study, were grown in DMEM media containing 5% fetal bovine serum (FBS), 100 IU/mL penicillin, and 100 μg/mL streptomycin. Human Embryonic Kidney 293 (HEK293A) cells, and all derivative cell lines, were grown in DMEM media containing 10% FBS, 100 IU/mL penicillin, and 100 μg/mL streptomycin. hTERT-BJ fibroblasts, and all derivative cell lines, were grown in DMEM:F12 media containing 10% FBS, 100 IU/mL penicillin, and 100 μg/mL streptomycin. $Braf^{V600E}/Pten^{-/-}$ mouse tumor cells (D4M.3 A) were a generous gift of Dr. Constance Brinckerhoff. D4M.3 A mouse tumor cells were maintained in DMEM:F12 media containing 5% FBS. Primary adult epidermal melanocytes were purchased from ATCC (PCS-200-013) and maintained in Dermal Cell Basal Medium (ATCC PCS-200-030) supplemented with an Adult Melanocyte Growth Kit (ATCC PCS-200-042). All FBS used in these studies was confirmed to either be naturally absent of tetracyclines or below 20 ng/mL by the manufacturer. All cells were maintained at 37 °C with 5% CO$_2$ atmosphere and maintained at sub-confluent levels for passaging and all experiments. Cultures were regularly checked for mycoplasma contamination utilizing a PCR detection kit (G238, ABM) or Hoechst staining. Bright-field images of tissue culture cells were captured on an Echo Revolve Hybrid Microscopy system at ×10 or ×20 (Echo Laboratories).

**Cell-line generation**. To generate the $BRAF^{V600E}$ doxycycline-inducible system Mel-ST, HEK293A, or hTERT-BJ fibroblasts were first infected with lentivirus generated from pLenti CMV TetR Blast (Tet Repressor) and selected. Following selection, cells were infected with lentivirus generated from pLenti CMV/TO $BRAF^{V600E}$ Neo or pLenti CMV/TO BRAF Neo, selected, and single-cell cloned to establish cell lines which demonstrated no basal expression and strong induction after doxycycline addition. The expression of $BRAF^{V600E}$ was confirmed using two different mutant-specific antibodies (VE1 and RM8 clones). To generate stably expressing H2B-GFP lines, cells were infected with lentivirus generated from pLenti H2B-GFP Blast. Mel-ST cells stably expressing empty vector (pLVX Puro), YAP-5SA (pBABE YAP-5SA Puro), and TAZ-4SA (pLVX Flag TAZ-4SA Puro) were generated via viral infection followed by selection. Tetraploid cells were generated by treating asynchronous cells with 4 μM DCB for 16 h, followed by gentle washing to remove drug (5 × 5 min); completion of cytokinesis was confirmed by phase-contrast imaging.

**Viral infections and transfections**. Mel-ST, HEK293A, or BJ Fibroblasts were infected for 12–16 h with virus-carrying genes of interest in the presence of 10 μg/mL polybrene, washed, and allowed to recover for 24 h before selection or single-cell cloning. Short-term viral infection of primary melanocytes was carried out

similarly, but with 2 μg/mL of polybrene. All RNAi transfections were performed using 25–50 nM siRNA with Lipofectamine RNAi MAX according to the manufacturer's instructions. Briefly, cells were seeded in 6 or 12-well plates either by addition of reverse transfection mixture overnight for 18 h, which was then washed with PBS and replaced with fresh media, or forward transfection mixture for 4 h, which was then replaced with fresh media. Cells were then incubated for 48–72 h prior to lysis at sub-confluent levels.

**Plasmid generation**. Plasmids encoding the Tetracycline Repressor, pLenti CMV TetR Blast (716-1), was a gift from Eric Campeau & Paul Kaufman (Addgene Plasmid #17492). To create pLenti CMV/TO BRAF Neo and pLenti CMV/TO BRAF$^{V600E}$ Neo, we performed Gateway cloning using Gateway LR Clonase II (Invitrogen) according to manufacturer instructions to insert BRAF or BRAF$^{V600E}$ into pLenti CMV/TO Neo DEST (685-3) using pENTR BRAF or pENTR BRAF$^{V600E}$. pENTR BRAF and pENTR BRAF$^{V600E}$ were gifts from Craig Ceol and pLenti CMV/TO Neo DEST (685-3) was a gift from Eric Campeau & Paul Kaufman (Addgene plasmid #17292). To generate pLenti CMV/TO NRAS$^{Q61R}$ Neo we performed Gateway cloning to insert NRAS$^{Q61R}$ from the donor vector pDONR223 NRAS$^{Q61R}$ into the destination vector pLenti CMV/TO Neo Dest. pDONR223 NRAS$^{Q61R}$ was a gift from Jesse Boehm, William Hahn, and David Root (Addgene plasmid #81652).

**Immunofluorescence and confocal microscopy**. Cells were plated on glass coverslips, treated as indicated, washed in 1× phosphate-buffered saline (PBS) (Boston Bioproducts), and fixed in 4% paraformaldehyde for 10 min. Cells were then washed in PBS-0.01% Triton X-100, extracted in PBS-0.2% Triton X-100 for 10 min, blocked in Tris-buffered saline (TBS)-bovine serum albumin (BSA) (10 mM Tris, pH 7.5, 150 mM NaCl, 5% bovine serum antigen, 0.2% sodium azide) for 1 h, and incubated with primary antibodies diluted in TBS-BSA for 1 h at RT or overnight at 4 °C in a humidified chamber. Primary antibodies were visualized using species-specific fluorescent secondary antibodies (Molecular Probes, Alexa Fluor secondaries, 488 nm, 568 nm, 1:500) and DNA was detected with 2.5 μg/mL Hoechst. F-actin was visualized using rhodamine-conjugated phalloidin (1:2000, Molecular Probes, R415). Immunofluorescence images for analysis were collected on a Nikon Ti-E inverted microscope equipped with a Zyla 4.2 PLUS (Andor) and X-Cite 120 LED light source at the same exposure. Confocal immunofluorescence images were collected on a Nikon Ti-E inverted microscope equipped with a C2 + laser scanning confocal head with 405 nm, 488 nm, 561 nm, 640 nm laser lines. Z-stacks were acquired with a series of 0.5–1 μm optical slices which were then converted into a single, max-intensity projected image. Images were analyzed using NIS-Elements Advanced Research (AR) and ImageJ (Version 1.51). To assess YAP localization, two small square regions of interest were drawn at random in individual cells with one in the nucleus, and one in the cytoplasm. The background-corrected, mean fluorescence intensity of YAP was subsequently measured in these regions of interest and a nuclear to cytoplasmic ratio was determined. To assess stress fiber quantity, images were background-corrected, contrast normalized, and then fibers obvious to the naked eye were counted.

**Live-cell imaging**. Cells stably expressing H2B-GFP were grown on glass-bottom 12-well tissue culture-treated dishes (Cellvis) and treated with drugs of interest. Immediately post-treatment imaging was performed on a Nikon Ti-E inverted microscope equipped with the Nikon Perfect Focus system. The microscope stage was enclosed within a temperature and atmosphere-controlled environment at 37 °C and 5% humidified CO$_2$. Fluorescent or bright-field images were captured every 5–10 min with an ×10 or 20×0.5 NA Plan Fluor objective at multiple locations for 72–96 h. All captured images were analyzed using NIS-Elements AR software. The mitotic length was calculated by counting the duration from nuclear envelope breakdown to anaphase onset.

**Tissue staining**. At the experimental endpoint mice were euthanized and mouse tumors or skin samples were dissected and immediately fixed in 4% paraformaldehyde (PFA) in PBS for 16 h at 4 °C. Tissues were then paraffin-embedded for sectioning and mounting. PFA-fixed paraffin-embedded tissue sections were cut at 5 μm and mounted onto positively charged coverslips (Colorfrost Plus, Thermo). Mounted tissue samples were deparaffinized using xylenes and rehydrated via an ethanol:water gradient. For hematoxylin and eosin staining (H&E), the tissues were incubated in hematoxylin (#14166, Cell Signaling Technology), rinsed in water, differentiated with 1% acid ethanol, blued with 0.1% sodium bicarbonate solution, rinsed in water, dehydrated, cleared, and mounted with Cytoseal XYL (Thermo Scientific). For immunohistochemistry (IHC) or immunofluorescence (IF), antigen unmasking was performed on rehydrated tissue sections using either a citric-acid retrieval buffer (Vector Labs) or Tris-EDTA retrieval buffer (10 mM Tris, 1 mM EDTA, 0.05% Tween-20). Heat-mediated antigen retrieval was performed using either a standard microwave (95 °C, 20 min) or Decloaking Chamber NxGen (Biocare Medical) (110 °C, 12 min), followed by cooling to room temperature (~30–60 min). Citric-acid retrieval was used for most antibodies; Tris-EDTA retrieval was used for MelanA antibodies. For IHC, tissue sections were washed, endogenous peroxidase activity was quenched using 3% hydrogen peroxide in PBS for 10 min, and then blocked for 1 h in 10% goat serum in TBS (Sigma-Aldrich).

For IF, tissue sections were blocked for 1 h in 10% goat serum in TBS. Following serum block, if necessary, tissue was incubated with Rodent Block M (Biocare Medical) for 30 min to block endogenous mouse IgG prior to murine primary antibody addition. Primary antibodies were diluted in 10% goat serum in TBS and incubated overnight at 4 °C in a humidified chamber. Following primary addition for IHC, slides were washed with TBS-0.01% Tween-20, incubated with anti-rabbit or mouse SignalStain Boost IHC detection reagent (Cell Signaling Technology) for 30 min and then developed with SignalStain DAB substrate kit (Cell Signaling Technology) according to manufacturer's instructions. Counterstaining was performed using hematoxylin, followed by dehydration, clearing, and mounting with Cytoseal XYL (Thermo Fisher). Images were captured at randomly selected points using a Nikon Ti-E inverted microscope equipped with a DS-Ri2 (Nikon). For IF, slides were incubated with species-specific fluorescent secondary antibodies (Molecular Probes) and 2.5 μg/mL Hoechst for 1 h at room temperature in a dark humidified chamber. Auto-fluorescence was quenched using Vector TrueVIEW according to the manufacturer's instructions, and slides were mounted using Prolong Gold Antifade (Invitrogen). Images were captured at randomly selected points using a Nikon Ti-E inverted microscope equipped with a Zyla 4.2 PLUS (Andor) and X-Cite 120 LED light source. For all staining experiments, tissue-specific secondary controls were included in each staining experiment to ensure specificity and control for endogenous tissue pigmentation levels.

**Protein extraction and immunoblotting**. Cells were gently washed twice in ice-cold 1X PBS and lysed using ice-cold cell lysis buffer (50 mM Tris-HCl, 2% w/v SDS, 10% glycerol) containing 1X HALT (dual-phosphatase and protease inhibitor, Thermo Fisher). Lysates were sonicated at 20% amplitude for 20 s, diluted in 4X Sample Buffer (Boston BioProducts), and resolved using SDS gel electrophoresis. For mouse tumor samples, 10 mg of tumor tissue was dissected and placed into RIPA lysis buffer (Boston BioProducts) supplemented with one Pierce protease inhibitor tablet and 1X HALT (Thermo Fisher). Tissue was mechanically dissociated initially using a 7-cm pestle (Kimble) followed by further homogenization with a 20-guage needle and brief sonication for 20 s at 20 kHz. Tissue lysate was then centrifuged at 4 °C and >15,000×g for 10 min. The supernatant was collected, diluted in 4× sample buffer to 2×, and resolved using SDS gel electrophoresis. For phos-tag immunoblots, a phos-tag gel was prepared with diluted Phos-Tag™ Acrylamide reagent (Wako Chemicals, AAL-107) according to the manufacturer's instructions utilizing manganese as the divalent cation. Proteins were then transferred onto PVDF membranes using a wet-tank transfer system (Bio-Rad), blocked for 1 h with TBS-0.1% Tween-20 containing 5% skim milk powder and then probed overnight at 4 °C with primary antibodies diluted in TBS-0.1% Tween-20 containing 1% skim milk powder. Bound antibodies were detected by incubating membranes with horseradish peroxidase-linked, species-specific, secondary antibodies (1:5000, Cell Signaling Technology) for 1 h, followed by 3 × 10 min washing in TBS-0.1% Tween-20, and then the addition of Clarity or Clarity Max ECL blotting substrate (Bio-Rad). Chemiluminescence acquisition was carried out using the Bio-Rad ChemiDox XRS + system and quantitative densitometry was measured using Image Lab (Bio-Rad).

**Human nevi and melanoma samples**. Human skin tissues with melanocytic nevi and melanoma were retrieved from archived material in the pathology laboratory at UMass Medical Center in compliance with all relevant ethical regulations and were determined to be exempt by the Institutional Review Board at UMass Medical Center (IRB H00007200). The tissue blocks were deidentified before tissue sections (10 μm) were obtained by the authors for IHC. These studies were reviewed in compliance with all relevant ethical regulations and were determined to be exempt by the Boston University School of Medicine Institutional Review Board (IRB H-37967).

**RNA isolation and qRT-PCR**. Total RNA from cultured cells was isolated using a Quick-RNA kit (Zymo Research). cDNA libraries were generated from RNA using the Superscript III kit and random hexamer primers (Invitrogen). Quantitative real-time PCR was performed using SYBR Green reagents in a StepOnePlus system (Applied Biosystems) according to manufacturer protocol. For each individual experiment, a technical triplicate was run which was then averaged to generate a single biological replicate. Primer sequences were as follows:
*CYR61*: forward, AGCCTCGCATCCTATACAACC; reverse, TTCTTTCACAAGGCGGCACTC
*AMOTL2*: forward, TTGGAATCTGCAAATCGCC; reverse, TGCTGTTCGTAGCTCTGAG
*GAPDH*: forward, GAGTCAACGGATTTGGTCG; reverse, CATTGATGGCAACAATATCCAC

**Antibodies**. The antibodies used herein categorized by technique and company.
Immunofluorescence (cells): Santa Cruz Biotechnology: YAP 63.7, 1:250 (detects both YAP/TAZ, sc-101199).
Immunofluorescence (tissue): Cell Signaling Technologies: YAP (1A12) #12395, 1:100; YAP/TAZ (D24E4) #8418, 1:100; YAP (D8H1X) #14074, 1:100. Abcam: gp100 (ab137078), 1:250; SOX10 (Rabbit, ab180862), 1:250; SOX10 (Mouse, ab216020), 1:250.

Immunohistochemistry: Cell Signaling Technologies: YAP/TAZ (D24E4) #8418, 1:100; YAP (D8H1X) #14074, 1:100; GFP (D5.1, cross reacts with YFP) #2956, 1:100; phospho-p44/42 ERK1/2 (Thr202/Tyr204) #9101, 1:100; Abcam: gp100 (ab137078), 1:250; SOX10 (Rabbit, ab180862), 1:250; SOX10 (Mouse, ab216020), 1:250; MelanA (ab210546), 1:500. Dako: S100 (IS504), pre-diluted by the manufacturer.

Immunoblotting: Cell Signaling Technologies: B-Raf (D9T6S), 1:1000; phospho-p44/42 ERK1/2 (Thr202/Tyr204) #9101, 1:1000; p44/42 MAPK (ERK1/2) #9102, 1:1000; RSK1/RSK2/RSK3 (3D27) #9355, 1:1000; phospho-p90RSK (Ser380) (D3H11) #11989, 1:1000; GAPDH (14C10) #2118, 1:1000; YAP (D8H1X) #14074, 1:1000; LATS1 (C66B5) #3477, 1:1000; phospho-LATS1 (Thr1079) (D57D3) #8654, 1:1000; TAZ (E8E9G) #83669, 1:1000; TAZ (V386) #4883, 1:1000; phospho-S6 (Ser235/236) (D57.2.2E) XP #4858, 1:1000; PTEN (138G6) #9559, 1:1000; p53 (1C12) #2524, 1:1000; phospho-Chk1 (Ser345) #2341, 1:500. Abcam: Vinculin (ab18058), 1:4000. Invitrogen: BRAF$^{V600E}$ (RM8 Clone) #MA5-24661, 1:1000-2000. Bethyl Laboratories: LATS1/LATS2 (A300-479A), 1:1000. Santa Cruz Biotechnology: Chk1 (G-4) sc-8408, 1:500; S6 (C-8) sc-74459, 1:1000; N-Ras (F155) sc-31, 1:1000. Spring Biosciences: BRAF$^{V600E}$ (VE1 Clone) # E19290, 1:1000. Cytoskeleton: RhoA (ARH05), 1:500.

**Soft-agar assays.** In six-well dishes, sterile 2% noble agar stock solution in water was dissolved by heating to ~40–45 °C, mixed with warm media, plated at a final concentration of 0.6%, and allowed to cool and solidify at 4 °C. Following solidification, gels were warmed back to 37 °C. Next, cells were trypsinized, counted, $1 \times 10^4$ cells were plated in 0.3% noble agar and allowed to solidify at room temperature or briefly at 4 °C. Plates were maintained in a cell culture incubator for 2–4 weeks with feedings of 1.5 mL of 0.3% agarose solution weekly. All drugs were maintained at 2× concentration in underlays and independent experiments were done in technical triplicate. Total colonies per well were counted using phase-contrast imaging on an Echo Revolve Hybrid Microscopy system at ×10 (Echo Laboratories). For imaging, gels were stained for 20 min with 0.1% Crystal Violet, gently washed multiple times overnight, and imaged on a Chemi-Doc XRS + system under Fast Blast with 0.015-0.1 sec exposure.

**EdU assays.** Mel-ST cells were seeded on coverslips a day before each collection at a density of $4.5 \times 10^4$ cells/well in a 12-well culture dish. Cells were treated with DMSO as a negative control or doxycycline to induce expression of BRAF$^{V600E}$. EdU was added for a 30-min pulse at a final concentration of 10 µM and cells were then fixed with 4% PFA at different timepoints since the addition of doxycycline. Incorporated EdU was visualized using the Click-iT EdU kit from Invitrogen according to the manufacturer's protocol (C10337), imaged via fluorescence microscopy on a Nikon Ti-E inverted microscope equipped with a Zyla 4.2 PLUS (Andor) and X-Cite 120 LED light source at the same exposure, and analyzed in ImageJ (version 1.51).

**RhoA pull-down activation assay.** Active, GTP-bound RhoA was assessed using the RhoA activation assay, bead pull-down format, from Cytoskeleton, Inc (BK036) according to manufacturer instructions. Briefly, dox-inducible BRAF$^{V600E}$ expressing Mel-ST cells were plated at $3 \times 10^4$ cells in six-well dishes and allowed to grow for 2–3 days. Prior to reaching 70% confluence, cells were serum-starved and treated ± doxycycline for 24 h, then stimulated with complete media for 6 h prior to ice-cold lysis. Following lysis, input was isolated from total lysate prior to flash freezing with liquid nitrogen and storage at −80 °C. Active RhoA pulldown was performed according to manufacturer instructions utilizing GST-tagged Rhotekin-RBD protein on agarose beads. Samples were then analyzed using immunoblot according to manufacturer protocols.

**Drug treatments.** The concentrations used for the MAPK pathway inhibitors (MEKi-1/2 and ERKi) were determined experimentally to be the doses at which phosphorylation of ERK1/2 and RSK1/2/3 returned to baseline despite the presence of oncogenic *BRAF* expression. The reagents used in these studies are as follows: MEKi-1: U0126 (Selleck Chemicals), 10 µM; MEKi-2: Trametinib (GSK1120212), 20 nM (Selleck Chemicals); ERKi: SCH772984, 20 nM (Selleck Chemicals); Hydroxyurea, 1 mM (Selleck Chemicals); doxycycline, 1 µg/mL (Sigma-Aldrich D9891); thymidine, 2.5 mM (Sigma-Aldrich T1895); RO-3306, 7 µM (Sigma-Aldrich SML0569)

**Population doubling and viability assays.** For population doubling assays, initially $1 \times 10^5$ cells ($c_{initial}$) were plated in 10-cm dishes. After 4 days of growth cells were trypsinized, counted, and $1 \times 10^5$ cells were plated again. Cells were trypsinized and counted again after 4 more days. The number of population doublings (pd) were calculated by inputting the counted number of cells ($c_{final}$) into the following equation: $pd = \log_2(\frac{c_{final}}{c_{initial}})$. For viability assays, $4 \times 10^3$ cells per well were plated into a white-bottom 96-well dish in technical duplicate to quintuplicate, dependent upon the experiment. Cells were then treated with indicated siRNA and/or drugs for 4 days. The 96-well plate was then allowed to equilibrate at room temperature for 30 min followed by the addition of Cell Titer Glo (Promega) to each well according to the manufacturer's instructions. Viability was then assessed

via luminescence using a BMG microplate reader and analyzed using BMG Labtech Optima v2.0R2 software. All technical replicates were averaged to generate a single biologic replicate.

**Copy-number analysis.** Log$_2$ copy-number values and/or putative copy-number GISTIC 2.0 values for *LATS1* and *LATS2* were downloaded from cBioPortal for the TCGA skin cutaneous melanoma (SKCM) subset of the PanCancer Atlas and Memorial Sloan Kettering Cancer Center (MSKCC), NEJM 2014 datasets. SKCM abbreviation represents skin cutaneous melanoma. Mutational status for *BRAF* and *NRAS* from the TCGA-SKCM dataset was also obtained from cBioPortal. Graphs were created in Prism 9.

**Copy-number analysis by ultra-low-pass whole-genome sequencing.** At the experimental endpoint, mice were euthanized and mouse tumors or skin samples were dissected, and immediately flash-frozen in liquid nitrogen and stored at −80 °C. For genomic DNA isolation, a Monarch Genomic DNA purification kit was utilized following the manufacturer's instructions. In brief, tumor or skin samples on ice were mixed with lysis buffer and proteinase K, vortexed, then incubated at 56 °C in a thermal mixer set to 1400 rpm agitation for 60 minutes. Lysed samples were then centrifuged for 3 min at >12,000×g. The supernatant was then mixed with RNase A, incubated for 5 minutes at 56 °C with full-speed agitation and then bound to the gDNA purification according to manufacturer instructions. gDNA was then eluted and stored at −20 °C until delivery to the Molecular Biology Core Facilities (MBCF) at Dana-Farber Cancer Institute (DFCI).

*Library preparation and sequencing.* gDNA was fragmented to 200 bp on a Covaris M220 instrument according to the manufacturer's protocol. Libraries were prepared using Swift S2 Acel reagents on a Beckman Coulter Biomek i7 liquid handling platform from approximately 200 ng of DNA according to the manufacturer's protocol with 14 cycles of PCR amplification. Finished libraries were quantified by Qubit fluorometer and fragment size distribution was evaluated by Agilent TapeStation 2200. Library pools were further evaluated for quality and balance with shallow sequencing on an Illumina MiSeq. Subsequently, libraries were sequenced targeting a depth of ~1X genome coverage on an Illumina NovaSeq6000 with paired-end 150 bp reads by the Molecular Biology Core Facilities at Dana-Farber Cancer Institute.

*Copy-number analysis.* Raw sequencing data (FASTQ files) were aligned to human and mouse reference genomes, GRCh37 and MM10 respectively, using BWA MEM (v.0.7.12). Both human and mouse alignment files were sorted by read names using Samtools (version 1.9). Software package Disambiguate (v. 1.0) was applied to sorted alignment files to infer the most likely source of each sequenced read (human or mouse genomes). Disambiguate output (BAM files) that comprised deconvoluted or "cleaned" human reads was used for further analysis. Pre-computed bin annotations for human genome (GRCh37) and bin sizes of 100 Kb, 500 Kb, and 1 Mb were obtained with QDNASeq package (v. 1.28.0), along with GC-content and mappability profiles required for data normalization and bias correction. Normalized copy-ratios were estimated for each deconvoluted alignment file using QDNASeq capabilities. CBS algorithm as implemented in DNA-copy package was used to produce genome segmentation profiles. CNVKit package was used to generate ideogram representations of per-chromosome copy-number profiles.

**Gene expression analysis.** Publicly available microarray dataset GSE61750 was obtained from the Gene Expression Omnibus (GEO) (https://www.ncbi.nlm.nih.gov/geo/query/acc.cgi?acc=gse61750). For GSEA, indicated groups were normalized via meandiv, probes collapsed to max probe, and a weighted enrichment metric (log$_2$ Ratio of Classes) was utilized with 1000 permutations to determine enrichment in indicated gene sets. The YAP/TAZ target score gene set for VAM and GSEA was derived from ref. [36] and ref. [37]. Gene set can be found in Supplementary Table 1. The Hippo component gene set was derived from Reactome with additional genes added and can be found in Supplementary Table 2.

**Single-cell RNAseq analysis.** Count matrix of GSE154679 was downloaded from the GEO repository (https://www.ncbi.nlm.nih.gov/geo/query/acc.cgi?acc=GSE154679)[25]. Cells that had less than 200 expressed genes, more than 4000 expressed genes or >17% of mitochondrial gene were removed from analysis. A total of 35,214 cells were retained for downstream analysis. Normalization, dimensionality reduction, cell clustering and data visualization were analyzed with Seurat package[81]. PCA dimension reduction was performed using top 2000 highest variance genes. The top 15 principal components were utilized to calculate the k-nearest neighbors of each cell. Cell clusters were determined using Louvain algorithm at a resolution = 0.5, which was high enough to obtain clusters associated with cell lineage identity. We used UMAP to visualize and confirm cell clustering. Melanocyte cell population was identified based on the expression of canonical gene markers. To identify subclusters within melanocyte cell populations, we re-analyzed cells within melanocyte cell clusters using the same analysis pipeline described above. To quantify gene sets activity in each cell, gene set testing of scRNA-seq data was performed following VAM pipeline[35]. Data Normalization

and highest variable genes were analyzed using pipeline described above. VAM method was executed using vamForSeurat() function on the melanocyte cell population for each gene set. Graphs were created in Seurat or data was imported into Prism 9 for graph creation.

**Murine studies.** All animal experiments were conducted according to protocols approved by the Institutional Animal Care and Use Committee (IACUC) at Boston University (Protocol # PROTO201800236). Per Boston University IACUC protocol the maximum allowable mouse tumor size is 20 mm in diameter which was not exceeded. For mouse studies, a mix of male and female mice of the C57BL/6 strain were utilized under standard housing conditions (temperature: 68–79 °F, humidity: 30–70%, light cycles: 12 h on, 12 h off). All other mouse strains were sourced from Jackson Laboratory: *Tyr::CreER^T2* (Jax # 012328), *Lats1^f/f* (Jax # 024941), *Lats2^f/f* (Jax # 025428), *BRAF^V600E* (Jax # 017837), *R26-YFP^LSL* (Jax # 006148) and were crossed in-house. 4-hydroxytamoxifen (4-HT [Sigma, H7904]) was dissolved in methanol to a concentration of 5 mg/ml. To induce knockout in *Lats1/2^−/−*, and *Braf^V600E/Lats1/2^−/−*, mice aged 8–12 weeks had an ~2 cm² shaved on the right flank using surgical clippers and sufficient 4-HT solution to wet the skin was applied once daily to the shaved area for 3 consecutive days. For *Braf^V600E* mice, a higher concentration of 4-HT was required; 4-HT was dissolved in dimethylsulf-oxide (DMSO) to a concentration of 25 mg/ml, and topical administration was performed as above. For the depilation experiments using *Lats1/2^−/−* mice, a 2 cm² was shaved on both the left and right flank of each mouse, and 5 mg/ml 4-HT was topically applied to each square for 3 consecutive days as above on the right flank, as well as on an approximately 2 cm length of the tail for 3 consecutive days. Two days after the final application, chemical depilation was performed by application of Nair hair remover onto the right flank for 15 s, followed immediately by wiping with a damp tissue to prevent irritation. For systemic knockout experiments, mice were injected with 20 mg/ml tamoxifen (Sigma, T5648) dissolved in corn oil to induce deletion in all Tyrosinase expressing cells. 8–12-week-old mice were injected daily with 100 µl tamoxifen for 5 consecutive days. Tumor development was measured weekly using calipers.

**Zebrafish studies.** Zebrafish were handled in accordance with protocols approved by the University of Massachusetts Medical School IACUC (Protocol A-2171). Constructs *Pmitfa:EGFP:pA* and *Pmitfa:YAP-5SA:pA* were used in the miniCoopR assay as previously described[65]. Briefly, *mitfa(lf)* mutant animals were bred, and single-cell stage embryos were injected with 25 pg of a single construct and 25 pg of Tol2 transposase mRNA. Successful Tol2-mediated integration of the construct into the genome rescued the *mitfa(lf)* phenotype, enabling melanocyte develop-ment. Equal numbers of male and female zebrafish were used. Melanocyte rescue was scored at 4–5 days of development, and rescued animals were grown to adulthood and monitored weekly for the presence of melanomas.

**Statistical analysis.** All quantitative data are presented as mean ± SEM, unless otherwise indicated. The number of samples (n) represents the number of biologic replicates or animals in study, unless otherwise indicated. Prism 9 was used for all statistical analyses and for the creation of most graphs.

**RNAi sequences.** The siRNA's used in this study are as follows:
Non-targeting #1 (Control siRNA) (Dharmacon):
UGGUUUACAUGUCGACUAA
LATS1 ON-TARGETplus SMARTpool (Dharmacon):
GGUGAAGUCUGUCUAGCAA; UAGCAUGGAUUUCAGUAAU
GGUAGUUGCUAUAUUAU; GAAUGGUACUGGACAAACU
LATS2 ON-TARGETplus SMARTpool (Dharmacon):
GCACGCAUUUUACGAAUUC; ACACUCACCUCGCCCAAUA
AAUCAGAUAUUCCUUGUUG; GAAGUGAACCGGCAAAUGC
YAP1 ON-TARGETplus SMARTpool (Dharmacon)
GCACCUACACUCUCGAGA; UGAGAACAAUGACGACCAA
GGUCAGAGAUACUUCUUAA; CCACCAAGCUAGAUAAAGA
WWTR1 ON-TARGETplus SMARTpool (Dharmacon)
CCGCAGGGCUCAUGAGUAU; GGACAAACACCCAUGAACA
AGGAACAAACGUUGACUUA; CCAAAUCUCGUGAUGAAUC.

**Reporting summary.** Further information on research design is available in the Nature Research Reporting Summary linked to this article.

## Data availability

TCGA and MSKCC datasets used in the study are publicly available and can be obtained from https://portal.gdc.cancer.gov. Gene set data used in the study are publicly available on the GEO repository: GSE61750, GSE154679. Sequencing data generated in this study are publicly available on an NCBI database, the Sequence Read Archive (SRA), accession code PRJNA835203. The authors declare that all other data supporting this study are either available within the article, supplementary figures, or available from the authors upon request. Source data are provided with this paper.

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

## Acknowledgements

We would like to thank Jackie King and Ross King for their unwavering resolve to promote melanoma awareness and research, as well Qi Sun, Shuyang Chen, and the entire Ganem Lab for wisdom and advice. We would also like to thank April Deng, Karen Dresser, Constance Brinckerhoff, Robert Weinberg, and Arthur Lander for sharing cell lines, reagents, and/or assistance. The results published herein are partly based upon data generated by the TCGA Research Network: https://www.cancer.gov/tcga. We thank all the patients who donated specimens to both the TCGA and the MSKCC database and the Molecular Biology Core Facilites at the Dana-Farber Cancer Institue for their assistance with the copy-number analysis. M.A.V. is supported by a Ruth L. Kirchstein National Research Service Award (F30) from the NCI (1F30CA228388) and was previously supported by a training grant from the NIGMS (5T32GM008541). E.X. and S.M. were both supported by an award from the Boston University Undergraduate Research Opportunities Program. X.V. is supported by the NHLBI (R01HL124392) and the American Cancer Society - Ellison New England Research Scholar Grant (RSG-17-138-01-CSM). N.M.K. was supported by the NHLBI (F31HL146163). C.J.C. was supported by NIAMS (AR063850) and by the USA Department of Defense (W81XWH2010288). N.J.G. is a member of the Shamim and Ashraf Dahod Breast Cancer Research Laboratories and this work was supported in his lab by the NIGMS (GM117150), the Harry J. Lloyd Charitable Trust, the Jackie King Young Investigator Award from the Melanoma Research Alliance, and the Searle's Scholars Program. This work was also supported by a pilot grant from the ACS and the Boston University Clinical and Translational Science Institute Bioinformatics Group (1UL1TR001430).

## Author contributions

M.A.V. and N.J.G. conceptualized the study, designed the in vitro experiments, and wrote the manuscript. M.A.V., N.K., X.V., and N.J.G. designed all in vivo studies. M.A.V. performed most of the cell biological assays, tissue staining, and imaging analysis. N.K. performed in vivo experiments, with the assistance of M.A.V. and K.K., and prepared all tissue samples. K.K., N.K., E.X., S.M., and A.T.L. assisted M.A.V. with the cell biological

and tissue assays. X.X. provided dermatopathology consult and tissue analysis. R.D. and C.C. performed zebrafish experiments. R.H. and J.D.C. completed the single-cell analysis and M.A.V. performed GSEA. L.H. and D.L. provided critical reagents.

## Competing interests

The authors declare no competing interests.
