## [Peer Review File · Nature Communications]

Inactivation of the hippo tumor suppressor pathway promotes melanomaREVIEWER COMMENTS

Reviewer #1 (Remarks to the Author):

In "Inactivation of the Hippo Tumor Suppressor Pathway Promotes Melanoma" by Vittoria et al, the authors find that inactivation of the YAP/TAZ signaling can play a determinative role in controlling BRAF-activated human melanocyte growth and melanoma formation in mice and zebrafish models, bringing a greater potential mechanistic understanding to the role of the Hippo in human melanoma. The experiments span an impressive array of cell culture and in vivo models and combine molecular genetics/scRNA analysis, biochemistry/intracellular signaling approaches, and multiple animal models to develop their largely well-supported conclusions. Overall, this manuscript would appear to be of broad interest to the cancer biology, cell biology, and melanoma fields, and is appropriate for publication with a few points addressed:

Major points:

- The authors are generally appropriately measured in their comments regarding the generalizability of the role of the Hippo pathway in melanoma formation – alteration of this pathway is a way in which melanocytes can circumvent growth arrest/growth limiting signals to produce a tumor. The ability of LATS 1 and 2 homozygous or compound heterozygous LOF in melanocytes to produce tumors is very interesting (Fig 5G), and the paper would be strengthened by a greater understanding of these tumors, or a softening of the description of these as melanomas (Discussion, page 9, 2nd to last paragraph). The authors show co-occurrence of LATS 1 and 2 loss in human melanoma (Fig 5A, which, to be fair, overrepresents Stage III/IV nodal mets in the TCGA dataset) and in combination with the mice/zebrafish data, provide a nice mechanistic explanation (including evidence that the effect is not mediated through p53). LATS 1 and 2 LOF leads to pERK accumulation, presumably in the absence of BRAF/RAS genetic alterations in the model, but it does not seem that naturally occurring human melanomas arise solely from LATS and/or Hippo dysfunction. Thus, if the authors have additional data to support these overgrowths as melanoma tumors (transplantable? Global gene expression from RNA-Seq of LATS-LOF induced tumor that is consistent with melanoma?), this could be included and further support that these are essentially melanomas. Alternatively, while these tumors do have histological and lineage markers (e.g. sox10) consistent with melanoma, the authors should highlight this point more – the tumors arise from melanocytes in a genetically engineered model that partially replicates features (LATS 1/2 LOF) of some human melanomas (still interesting and of great value, but better contextualization).
- In Figure 4 A, expression of dct is highlighted in the scRNA data (and Mlana Figure S4). Given the presence of different melanocyte subtypes/clusters that are identifiable, it may be of value to include sox10, mitf, tyr expression in accompanying UMAP plots to further describe the differentiation state/fidelity of melanocyte marker gene expression.

Minor points:

- Images of the zebrafish tumors (Fig 6E), either photos of gross tumors and/or histological images, should be included to show the appearance of these tumors.
- If available, images of normal adjacent tissue (Figure 5D, G) would help the non-derm specialist better visualize the degree of staining/overactivation of provided melanoma markers/signaling pathway indicators (pERK1/2) in the spontaneous and induced tumors.
- In the discussion on page 9, the authors note frequent GNAQ/11 mutations (and the linkage to YAP/TAZ) in >80% of uveal melanomas, and more relevant likely to this paper, in ~6% of cutaneous melanomas. Uveal and cutaneous melanomas have distinct mutation signatures (as noted in the discussion, but also BAP1 loss in uveal, high UV-induced alterations in cutaneous melanoma) and quite different clinical behaviors (minimal response to immunotherapy for uveal melanoma, high propensity for liver-only mets in uveal melanoma). The concluding sentence of that paragraph would seem more accurate if softened or better clarified in determining that "YAP/TAZ-TEAD activation... relatively common in melanoma" based on the earlier comments in that paragraph.

Reviewer #3 (Remarks to the Author):

Summary: The study by Vittoria et al. emphasizes the tumor suppressive function of the Hippo pathway (inactivation of the YAP/TAZ transcriptional program), demonstrating that this pathway is an early shield in the defense against nevi-to-melanoma evolution, and that the pathway is responsible for halting mutated melanocyte growth and keeping nevi benign. While the majority of previous studies focusing on YAP/TAZ activation in melanoma have revolved around its function in drug resistance and melanoma progression a few have identified the very early role of YAP/TAZ activity in melanoma tumor initiation. The study is nevertheless novel, being the first to focus on YAP/TAZ activation in initial tumorigenesis specifically in cutaneous melanoma harboring the hyperactive BRAFV600E mutation. The study is technically diverse, with an appropriate model of non-transformed melanocyte behavior before and immediately after gaining the BRAFV600E mutation. The pitfalls of this study stem from the disconnected survey of Hippo-related processes that occur upon early oncogene hyperactivation, and thus the lack of a clear model or timeline of events that occur in benign nevi formation vs. melanoma tumor initiation (what truly activates the Hippo pathway vs. what is a consequence of Hippo signaling; and when the Hippo pathway succeeds vs. when it does not). While the work is strong overall, it will be essential to see the final outcomes of these processes illustrated in a schematic to enhance the discussion of how the observations in the manuscript all connect.

Major conceptual critiques:

1. Despite clear attempts to find the activating signal of the Hippo pathway upon BRAFV600E mutation gain, the authors did not find the true driver. While we applaud the inclusion of negative data, the negative data with mitotic slippage and the unconvincing data of cytoskeleton changes should be moved to the supplement to improve the manuscript flow and make space for more tests of potential upstream signaling events. The activation may be directly related to hyperactive MAPK signaling (Fig. 3A, B), but the data in their current state are correlational at best, since the MEKi and ERKi treatments are only surveyed at the 24 hr mark. Earlier effects of ablating the hyperactive MAPK signaling would be more convincing to show causality. One could also overexpress/inhibit adjacent RTK signaling to see if other similar pathways may be involved in Hippo regulation.
2. After moving the negative/weaker data to the supplement, the authors will have more space to showcase the MGH-CP1 drug that only appears for one experiment at the very end of the paper. Performing a few more in vitro experiments with this drug would not only enhance some signaling conclusions of the paper, but would also complement the elegant mouse work that has already been done to take the manuscript to a higher translational level.
3. Overall, the manuscript suffers from an organization issue and a disconnect between some of the data. The negative/unconvincing data about the potential Hippo pathway trigger (Figs. 2 and 3) fall short and distract from the interesting phenomena about early BRAF mutation gain and Hippo pathway activation, which is where we end up again when we arrive at Fig. 4 (except this time there is some nice in vivo context). Perhaps the Hippo initiation trigger can be hashed out in vitro and in vivo before the authors begin to tease out the triggering step (hopefully with some more tests in their revision). After finding the driver, their conceptual model will become clearer and they will be able to create an illustrative schematic of the tumor initiation phenomenon and MAPK/Hippo signaling changes that ensue when Hippo signaling is and is not present.

Data/experimental critiques:

Fig. 1 –To strengthen the use of the model cell line, it would be nice to have a simple growth curve (cell count over 7 days, for example) of MEL-ST with/without dox, and with/without LATS siRNA.

Fig. 2 – The time-lapse imaging here followed by the mitosis phenotype analysis is quite nice, but these data could be strengthened by a few additional simple cell-cycle experiments. For example, measuring EdU vs. Hoechst staining to get a quantitative read on cell-cycle phases after initial

BRAFV600E activation and DNA content/aneuploidy. Overall, while negative data are good to include, in this case they are distracting in a main figure.

Fig. 3 – This is the weakest figure by far. (Figs. 3A, 3B): It is hard to say that ERK activity is closely or directly responsible for LATS phosphorylation, since, if that were true, inhibiting MAPK activity would cause loss of phosphorylation on LATS or YAP in much shorter time scales (<2 hr), which are not tested here; MEK or ERK inhibitors are only tested at the 24 hr time point. It cannot be ruled out that hippo pathway inactivation is a consequence of drug-induced cell-cycle exit. (Figs. 3D, S3D, S3E): These data are not at all convincing. Images of phalloidin stain look exceptionally similar in the images before and after dox treatment. This stress fiber analysis is not convincing to show actin changes. Perhaps higher resolution images would help with arrows to demonstrate what is really being quantified in the supplementary plot Figure S3E. (Fig. S3F) 25-50 μ M of RAC1i seems like an extremely high dose, since you get p-YAP loss without dox as well. Can you rule out off target effects of those treatments? The ROCKi and RAC1i experiment is in general not convincing at identifying the BRAFV600E-induced hippo signaling. The trigger is clearly still unidentified.

Fig. 4 – This figure feels like it should arrive earlier following all the in vitro work with MEL-ST cells. Also why is histological immunostaining (Fig. 4G & Fig. S4H) only shown for benign nevi? Can you show YAP staining for at least one (matched somehow if possible) malignant nevus case?

Fig. 5 – Validation of Lats^{-/-} genotype is unclear. It would be nice to have the YFP incidence in melanocytes quantified for a population of cells (n>100) and western blots of isolated nevi from the mice. How much Lats is already gone even before tamoxifen treatment should also be shown. Additionally, it was mentioned that the BRAFV600E control alone did not induce spontaneous melanoma formation; where are those data? A timeline/overarching summary of each condition and the formation of the nevi and into melanoma formation would have been helpful. Is it possible to plot this with the existing datasets gathered? Testing overactivation of this pathway in vivo would also strengthen the story; such experiments should perhaps be discussed.

Fig. 6 – This figure is lackluster, especially following the impressive Fig. 5 data. Also, why was the MGH-CP1 drug used for one simple viability experiment at the end of the manuscript? Could this not be employed in other assays? As the authors write, "Given MAPK inhibitor resistance remains a significant component of treatment failure, our data suggests that co-targeting MAPK and YAP-TEAD signaling could simultaneously prevent resistance (69) and decrease melanoma cell viability."

Minor comments:

Need quantifications for all westerns, preferably in the main figures. Several are missing.
Need to change "u" to the proper micro symbol throughout plots in the manuscript.
Fig 6A resolution is extremely poor.

Reviewer #4 (Remarks to the Author):

In this manuscript, Vittoria et al. show that oncogenic BRAF activates the Hippo tumor suppressor pathway, which limits melanocyte proliferation through YAP/TAZ inactivation. In addition, they show that Lats1/2 deletion enables oncogenic BRAFV600E-expressing melanocytes to bypass nevus formation and rapidly form melanoma. Thus, the authors propose that Hippo pathway activation sustains nevus melanocyte arrest and prevents melanoma development. The conclusions follow from the presented data and the findings are interesting and add a new perspective to the Hippo pathway in oncogenesis. However, a number of points need to be addressed:

1. Fig1C & D show that YAP is not fully inactive in BRAFV600E expressing cells. In Fig1D there are no noticeable differences in terms of YAP nuclear localization between control and +Dox. In contrast, BRAFV600E expressing cells show strongly decreased expression of Cyr61 and Amol2. In

order to exclude that the expression of Cyr61 and Amol2 is driven by factors other than YAP/TAZ, the authors should test to what degree Yap/Taz knockdown inhibits Cyr61 and Amol2 expression. In addition, this could be done on top of the BRAFV600E mutation to test whether the effect of BRAFV600E on Cyr61 and Amotl2 requires Yap/Taz.

2. The observation that BRAFV600E impairs mitosis and can potentially explain the formation of multinucleated melanocytes in human nevi is interesting. However, the authors failed to show such mechanism in vivo in mice and cite unpublished data in zebrafish. What happens to the ploidy of mouse or human melanocytes expressing BRAFV600E? Can YAP-5SA rescue the polyploidy phenotype? If YAP can rescue the decreased proliferation phenotype of BRAF expressing melanocytes, what is the ploidy of the cells that respond to YAP? Can BRAF polyploid cells proliferate in response to YAP-5SA?

3. Fig3: The effects of BRAFV600E overexpression on stress fiber formation is not obvious as shown by the similar phalloidin staining (Fig3D). Also, the effects of ROCKi and RAC1i are indistinguishable in control or BRAFV600E expressing cells. I am convinced that there really is an effect, very underwhelming.

4. It is difficult to appreciate tumor load in the absence of pictures showing whole tissues and tumors. The authors should provide pictures of Lats KO - B BRAFV600E mice (depilated/non-depilated) with tumors.

5. Why was zebrafish used to induce tumors with YAP-5SA in melanocytes when mice would be preferable?

6. Discussion: "Questions remain as to how Hippo pathway activation so strongly promotes melanoma development" should rather be "Hippo pathway inactivation"?

Responses to Reviewer #1

“The experiments span an impressive array of cell culture and in vivo models and combine molecular genetics/scRNA analysis, biochemistry/intracellular signaling approaches, and multiple animal models to develop their largely well-supported conclusions. Overall, this manuscript would appear to be of broad interest to the cancer biology, cell biology, and melanoma fields, and is appropriate for publication with a few points addressed”.

We are appreciative of the reviewer’s positive comments, as well as for their overall thorough and thoughtful review.

“The authors are generally appropriately measured in their comments regarding the generalizability of the role of the Hippo pathway in melanoma formation – alteration of this pathway is a way in which melanocytes can circumvent growth arrest/growth limiting signals to produce a tumor. The ability of LATS 1 and 2 homozygous or compound heterozygous LOF in melanocytes to produce tumors is very interesting (Fig 5G), and the paper would be strengthened by a greater understanding of these tumors, or a softening of the description of these as melanomas (Discussion, page 9, 2nd to last paragraph). The authors show co-occurrence of LATS 1 and 2 loss in human melanoma (Fig 5A, which, to be fair, overrepresents Stage III/IV nodal mets in the TCGA dataset) and in combination with the mice/zebrafish data, provide a nice mechanistic explanation (including evidence that the effect is not mediated through p53). LATS 1 and 2 LOF leads to pERK accumulation, presumably in the absence of BRAF/RAS genetic alterations in the model, but it does not seem that naturally occurring human melanomas arise solely from LATS and/or Hippo dysfunction. Thus, if the authors have additional data to support these overgrowths as melanoma tumors (transplantable? Global gene expression from RNA-Seq of LATS-LOF induced tumor that is consistent with melanoma?), this could be included and further support that these are essentially melanomas. Alternatively, while these tumors do have histological and lineage markers (e.g. sox10) consistent with melanoma, the authors should highlight this point more – the tumors arise from melanocytes in a genetically engineered model that partially replicates features (LATS 1/2 LOF) of some human melanomas (still interesting and of great value, but better contextualization).”

We agree with the reviewer’s general sentiment that the manuscript would be improved with better contextualization of our tumor findings. In the discussion we now emphasize the point that our mouse model only partially replicates features of human melanoma. We also further discuss how the *Lats1/2*^{-/-} murine tumors do not initially form pigmented growths, which is different from other murine melanoma models which initially exhibit pigmented lesions before undergoing dedifferentiation and losing pigmentation as they transition to the melanoma invasive state. We have added the following key sentences to the manuscript:

“Co-heterozygous loss of LATS1/2 and amplification of YAP1 are observed in primary and metastatic human melanoma (31, 32, 38). However, it should be noted that functional impairment of the Hippo pathway alone is not observed in human melanoma, and thus our model only partially replicates features of human tumors.”

“It is believed that activating MAPK mutations are critical for human melanocyte transformation; however, we found that murine melanocytes lacking *Lats1/2* rapidly developed into melanomas without the initial presence of any other genetic alterations that stimulate the MAPK pathway. Questions therefore remain as to how Hippo pathway inactivation alone can so strongly promote melanoma development in mice...”

“While ~30-40% of all human melanomas, and the vast majority of human metastatic melanomas, demonstrate evidence of WGD (78), our mouse melanoma models failed to exhibit this genomic feature (Figure S8). This is likely because the mouse models are already potently tumorigenic without the need for the additional oncogenic effects imparted by a WGD; nevertheless, the lack of any appreciable WGD in murine melanomas illustrates that these models have limitations in recapitulating the human disease (45, 76-78).”

“In Figure 4 A, expression of *dct* is highlighted in the scRNA data (and *Mlana* Figure S4). Given the presence of different melanocyte subtypes/clusters that are identifiable, it may be of value to include

sox10, mitf, tyr expression in accompanying UMAP plots to further describe the differentiation state/fidelity of melanocyte marker gene expression.”

We have now revised this figure to include expression data for *Sox10*, *Mitf*, and *Tyr* as per the reviewer's suggestion. As expected, the expression of these genes was exclusively enriched in the subpopulation of cells previously identified as melanocytes using *Mlana* and *Dct*. Interestingly, further unsupervised sub-clustering of the melanocytes revealed that cluster 3 contained the highest expression of tyrosinase. This finding provides further evidence that cluster 3 represents follicular melanocytes as suggested in our manuscript and in alignment with the original analysis in Ruiz-Vega et al. Clusters 0, and 1, which represent nevus melanocytes, have strong expression of *Mitf* and *Tyr* as expected. These data are now included in Supplementary Figure 2.

“Images of the zebrafish tumors (Fig 6E), either photos of gross tumors and/or histological images, should be included to show the appearance of these tumors.”

We collected images of the gross pigmented tumors induced by constitutively active YAP in melanocytes. A representative image is now included as Figure 6E.

“If available, images of normal adjacent tissue (Figure 5D, G) would help the non-derm specialist better visualize the degree of staining/overactivation of provided melanoma markers/signaling pathway indicators (pERK1/2) in the spontaneous and induced tumors.”

This is an excellent suggestion and we have now included images of SOX10, YAP/TAZ, and p-ERK1/2 staining in normal tissue adjacent to tumors induced by *Lats1/2* deletion. Additionally, we included images of murine skin that was induced with tamoxifen but had not yet formed tumors, allowing for better appreciation of the staining pattern of the normal tissue prior to tumorigenesis. These images are now shown in Supplementary Figure 6.

“In the discussion on page 9, the authors note frequent GNAQ/11 mutations (and the linkage to YAP/TAZ) in >80% of uveal melanomas, and more relevant likely to this paper, in ~6% of cutaneous melanomas. Uveal and cutaneous melanomas have distinct mutation signatures (as noted in the the discussion, but also BAP1 loss in uveal, high UV-induced alterations in cutaneous melanoma) and quite different clinical behaviors (minimal response to immunotherapy for uveal melanoma, high propensity for liver-only mets in uveal melanoma). The concluding sentence of that paragraph would seem more accurate if softened or better clarified in determining that “YAP/TAZ-TEAD activation... relatively common in melanoma” based on the earlier comments in that paragraph.”

We agree with the reviewer on the lack of clarity. The main point we wanted to convey was that Hippo pathway inactivation occurs in human melanomas, and that this inactivation can arise through different mechanisms. In the revised manuscript, we now show a panel of human melanoma samples stained for YAP to demonstrate that some melanoma tumors do exhibit strong nuclear YAP localization, indicative of Hippo pathway inactivation (Figures 3G and S3E) (31, 38). We removed the statement that “YAP/TAZ-TEAD activation appears to be relatively common in melanoma”, as this wording was vague. We now emphasize that there may be multiple routes to Hippo pathway inactivation in melanomas aside from *LATS1/2* loss, with mutations in *GNAQ/GNA1* proteins being one such way. We now write:

“Moving forward, it will be important to define additional mechanisms by which human melanoma cells circumvent the Hippo pathway to activate YAP (38, 70, 71). For example, >80% of uveal melanomas and ~6% of cutaneous melanomas have mutations in GNAQ/GNA1 proteins, which stimulate RhoA activity and activate YAP/TAZ independent of any alterations in LATS1/2 (72-74).”

Responses to Reviewer #2

“The study is nevertheless novel, being the first to focus on YAP/TAZ activation in initial tumorigenesis specifically in cutaneous melanoma harboring the hyperactive BRAFV600E mutation. The study is technically diverse, with an appropriate model of non-transformed melanocyte behavior before and immediately after gaining the BRAFV600E mutation.”

We are happy that the reviewer appreciates the novelty of the study and our use of multiple technical approaches, and we are grateful for his/her thoughtful comments about improving the overall organization of the manuscript.

“Despite clear attempts to find the activating signal of the Hippo pathway upon BRAFV600E mutation gain, the authors did not find the true driver.”

Since our initial submission, we have continued to search for the signal that activates the Hippo pathway upon expression of oncogenic *BRAF*. To that end, we have now acquired new data demonstrating that activation of MAPK leads to significant reductions in RhoA activity in melanocytes *in vitro*. Such reductions in RhoA are well known to activate LATS1/2 and lead to Hippo pathway activation. We have included these data in Figure 4F. It should also be noted that an accompanying manuscript from Darp et al. independently demonstrates that activation of oncogenic *BRAF* reduces RhoA activity in other cell types. We have included this manuscript with this submission.

“While we applaud the inclusion of negative data, the negative data with mitotic slippage and the unconvincing data of cytoskeleton changes should be moved to the supplement to improve the manuscript flow and make space for more tests of potential upstream signaling events.”

We agree that the immunofluorescence images of cytoskeletal changes, despite our efforts to quantify them in an unbiased way, are not fully convincing and distract from the main message of our manuscript. We have now moved that data to Supplementary Figure 5. We do wish to keep the mitotic slippage data, as we feel that the abnormal mitosis, mitotic slippage, and subsequent whole genome-doubling induced by oncogenic *BRAF* is both significant and currently unappreciated by the cancer biology field. It is now recognized that WGD is very common in many solid tumors, especially melanoma, and so we would like to highlight a mechanism by which oncogenic *BRAF* may induce an initial WGD. However, we have significantly de-emphasized this point in the revised manuscript and these data are now introduced later in the manuscript to preserve the overall flow of the publication.

“The activation may be directly related to hyperactive MAPK signaling (Fig. 3A, B), but the data in their current state are correlational at best, since the MEKi and ERKi treatments are only surveyed at the 24 hr mark. Earlier effects of ablating the hyperactive MAPK signaling would be more convincing to show causality.”

This is an excellent point. We have now performed time course analyses to assess the timing of Hippo pathway activation following induction of oncogenic *BRAF* both with and without an ERK1/2 inhibitor. We find that expression of *BRAF* induces a steady increase in Hippo pathway activation from 2-24 h and that this activation is fully abrogated by an ERK1/2 inhibitor. These data support our conclusion that hyperactive MAPK signaling indirectly promotes Hippo pathway activation, at least in part, through downstream effects of cytoskeletal alterations. This is now included as Supplementary Figure 5A-B.

“One could also overexpress/inhibit adjacent RTK signaling to see if other similar pathways may be involved in Hippo regulation.”

This is a critical point. As activating mutations in *NRAS* are also common in melanoma (and stimulate MAPK pathway activation similar to RTK activation), we assessed Hippo pathway activation after inducible expression of *NRAS*^{Q61R} in melanocytes. We found that expression of oncogenic *NRAS* similarly activates the Hippo pathway (Supplementary Figure 1G).

“After moving the negative/weaker data to the supplement, the authors will have more space to showcase the MGH-CP1 drug that only appears for one experiment at the very end of the paper. Performing a few more in vitro experiments with this drug would not only enhance some signaling conclusions of the paper, but would also complement the elegant mouse work that has already been done to take the manuscript to a higher translational level.”

We agree that providing new evidence that YAP/TAZ may represent new therapeutic targets in melanoma would have important translational significance and that our prior evidence was limited. We examined whether YAP/TAZ inhibition could impair the viability of human melanoma cell lines. We found that inhibition of YAP/TAZ with the small molecule MGH-CP1 impaired the viability of 8 out of 9 human melanoma cell lines, with 4 out of 9 demonstrating significantly impaired viability over immortalized non-tumorigenic melanocytes. We include these data in Figure 6F. Together, these data suggest YAP and TAZ may represent important new therapeutic targets in human melanoma.

“Overall, the manuscript suffers from an organization issue and a disconnect between some of the data. The negative/unconvincing data about the potential Hippo pathway trigger (Figs. 2 and 3) fall short and distract from the interesting phenomena about early BRAF mutation gain and Hippo pathway activation, which is where we end up again when we arrive at Fig. 4 (except this time there is some nice in vivo context). Perhaps the Hippo initiation trigger can be hashed out in vitro and in vivo before the authors begin to tease out the triggering step (hopefully with some more tests in their revision). After finding the driver, their conceptual model will become clearer and they will be able to create an illustrative schematic of the tumor initiation phenomenon and MAPK/Hippo signaling changes that ensue when Hippo signaling is and is not present.”

We fully agree with these suggestions and have significantly reorganized the paper to address this concern. We have also included a schematic summary of our findings, which we think clarifies our major findings and model. This schematic is now in Figure 6.

“To strengthen the use of the model cell line, it would be nice to have a simple growth curve (cell count over 7 days, for example) of MEL-ST with/without dox, and with/without LATS siRNA.”

This is an excellent suggestion and we have now completed these population doubling assays. These data nicely demonstrate that oncogenic BRAF expression significantly impairs the growth of Mel-ST cells, but that inhibition of LATS1/2 using a potent and selective LATS kinase inhibitor completely rescues this growth defect. These data are now included in Figure 3A and 3C-D.

“Fig. 2 – The time-lapse imaging here followed by the mitosis phenotype analysis is quite nice, but these data could be strengthened by a few additional simple cell-cycle experiments. For example, measuring EdU vs. Hoechst staining to get a quantitative read on cell-cycle phases after initial BRAFV600E activation and DNA content/aneuploidy.”

We have taken the reviewer’s advice and performed Edu incorporation assays. These experiments demonstrate that Hippo pathway activation following expression of oncogenic BRAF leads to an overall decrease in the cells entering S-phase. This is included in Supplementary Figure 3B.

“Fig. 3 – This is the weakest figure by far. (Figs. 3A, 3B): It is hard to say that ERK activity is closely or directly responsible for LATS phosphorylation, since, if that were true, inhibiting MAPK activity would cause loss of phosphorylation on LATS or YAP in much shorter time scales (<2 hr), which are not tested here; MEK or ERK inhibitors are only tested at the 24 hr time point. It cannot be ruled out that hippo pathway inactivation is a consequence of drug-induced cell-cycle exit.”

We agree on these points, and as described above, we have now done a time course analyses to assess the timing of Hippo pathway activation following induction of oncogenic BRAF both with and without an ERK1/2 inhibitor. We find that expression of BRAF induces a steady increase in Hippo pathway activation from 2-24 hrs and that this is completely abrogated by an ERK1/2 inhibitor. These data are now included in Supplementary Figure 4B.

“Figs. 3D, S3D, S3E): These data are not at all convincing. Images of phalloidin stain look exceptionally similar in the images before and after dox treatment. This stress fiber analysis is not convincing to show actin changes. Perhaps higher resolution images would help with arrows to demonstrate what is really being quantified in the supplementary plot Figure S3E. (Fig. S3F).”

We agree that the effects on the actin cytoskeleton are hard to appreciate by just looking at a limited number of stained cells. For that reason, we developed an unbiased method to quantitate the effects on stress fibers to make our data more rigorous. Nevertheless, to better demonstrate the effect, we now include higher resolution photos of representative cells. In addition, we also include phase-contrast images of the cells, as the cytoskeletal changes induced by BRAF expression are more conspicuous when looking at total cell morphology. These changes are included in Supplementary Figures 3A and 5C-D.

“25-50 μ M of RAC1i seems like an extremely high dose, since you get p-YAP loss without dox as well. Can you rule out off target effects of those treatments? The ROCKi and RAC1i experiment is in general not convincing at identifying the BRAFV600E-induced hippo signaling. The trigger is clearly still unidentified.”

We agree that those small molecule inhibitors may have off-target effects and thus have removed them from this study. Our new data demonstrating that activation of MAPK leads to significant reductions in RhoA activity in melanocytes is more direct and convincing. We have included this significantly stronger data in Figure 4.

“Fig. 4 – This figure feels like it should arrive earlier following all the in vitro work with MEL-ST cells. Also why is histological immunostaining (Fig. 4G & Fig. S4H) only shown for benign nevi? Can you show YAP staining for at least one (matched somehow if possible) malignant nevus case?”

We agree that this work should come earlier, and we have now moved the RNAseq analysis and staining of nevi to Figure 2. We also agree it would be nice to include staining of intermediate lesions that lie in between a common nevus and melanoma *in situ*. Unfortunately, dysplastic nevi are difficult to identify with confidence and whether they represent true intermediary lesions remains heavily debated. We did not feel comfortable including this type of analysis with the samples we had. However, we now include YAP staining for several malignant melanoma samples isolated from patients. These data reveal that YAP is nuclear in a subset of human melanoma tissue samples. This is included in Figure 3 and Supplementary Figure 3.

“Fig. 5 – Validation of Lats-/- genotype is unclear. It would be nice to have the YFP incidence in melanocytes quantified for a population of cells ($n > 100$) and western blots of isolated nevi from the mice. How much Lats is already gone even before tamoxifen treatment should also be shown.”

These mouse models have previously been validated/published and every litter was genotyped prior to use in experiments. We now include immunoblots from tumors demonstrating loss of Lats1 protein. These data are included in Figure 6. While not included in this manuscript, we have also collected RNA sequencing data from tumor samples that confirm *Lats1/2* excision. Unfortunately, we were unable to perform western blots from nevi, as *Lats1/2*^{-/-} mice do not form nevi prior to tumorigenesis.

To assess the efficiency of Cre recombinase in mice, we performed IHC on hair follicle cells after painting with tamoxifen. Serial section staining revealed that essentially all mature melanocytes in hair follicles (as identified by GP100 stain) were YFP positive (as shown in Figure S6E).

We believe that calculating the exact percentage of *Lats1/2* loss prior to tamoxifen treatment would not change our overall conclusions, and that such studies would be technically challenging as no IHC-rated validated antibodies exist for *Lats1* or *Lats2*, and melanocytes are relatively rare and difficult to isolate for western blotting.

“Additionally, it was mentioned that the BRAFV600E control alone did not induce spontaneous melanoma formation; where are those data? A timeline/overarching summary of each condition and the formation of the nevi and into melanoma formation would have been helpful. Is it possible to plot this with the existing datasets gathered?”

This is a great suggestion. We have generated a table of all conditions and annotated nevus/melanoma formation timelines obtained from a sampling of previous studies. This is included in Supplementary Figure 6.

“Testing overactivation of this pathway in vivo would also strengthen the story; such experiments should perhaps be discussed.”

We agree it would be very interesting to examine how restoration/overactivation of the Hippo pathway activity modulates melanoma growth *in vivo*, but we feel that developing novel mouse models with inducible regulation of Hippo pathway activity is beyond the scope of this manuscript.

“This figure is lackluster, especially following the impressive Fig. 5 data. Also, why was the MGH-CP1 drug used for one simple viability experiment at the end of the manuscript? Could this not be employed in other assays?”

We agree. As mentioned above, we examined whether YAP/TAZ inhibition could impair the viability of human melanoma cell lines. We found that inhibition of YAP/TAZ with the small molecule MGH-CP1 impaired the viability 8 out of 9 human melanoma cell lines. We include these data in Figure 6.

“Need quantifications for all westerns, preferably in the main figures. Several are missing.”

We apologize for pushing some quantifications to the supplemental due to size constraints and not including other quantifications entirely despite having the data. We have now included all quantifications next to their corresponding western blots.

“Need to change “u” to the proper micro symbol throughout plots in the manuscript.”

We have made this formatting change.

“Fig 6A resolution is extremely poor.”

We have exported this image with higher-resolution.

Responses to Reviewer #3

“The conclusions follow from the presented data and the findings are interesting and add a new perspective to the Hippo pathway in oncogenesis.”

We thank the reviewer for his/her kind words and helpful suggestions to improve the manuscript.

“Fig1C &D show that YAP is not fully inactive in BRAFV600E expressing cells. In Fig1D there are no noticeable differences in terms of YAP nuclear localization between control and +Dox. In contrast, BRAFV600E expressing cells show strongly decreased expression of Cyr61 and Amol2. In order to exclude that the expression of Cyr61 and Amol2 is driven by factors other than YAP/TAZ, the authors should test to what degree Yap/Taz knockdown inhibits Cyr61 and Amol2 expression. In addition, this could be done on top of the BRAFV600E mutation to test whether the effect of BRAFV600E on Cyr61 and Amotl2 requires Yap/Taz.”

We have now performed qPCR analysis on cells treated with YAP/TAZ siRNA and demonstrated that inhibition of YAP/TAZ activity reduces *CYR61* and *AMOTL2* expression levels similar to oncogenic BRAF expression. Additionally, these data demonstrate that BRAF expression does not decrease *CYR61* and *AMOTL2* levels significantly further than RNAi-mediated knockdown of YAP/TAZ, suggesting that BRAF expression reduces *CYR61* and *AMOTL2* predominantly through Hippo pathway activation. These data are included in Supplementary Figure 1.

“The observation that BRAFV600E impairs mitosis and can potentially explain the formation of multinucleated melanocytes in human nevi is interesting. However, the authors failed to show such mechanism in vivo in mice and cite unpublished data in zebrafish.”

We attempted to quantify the fraction of multinucleated melanocytes in benign mouse nevi following expression of oncogenic BRAF. While multinucleated cells were observed, they were quite rare. Thus, while we show that oncogenic BRAF can strongly induced mitotic failure *in vitro*, it seems this effect is minimized in mice. However, an accompanying manuscript by Revati et al also in review at Nature Communications (which we have attached with these revisions) demonstrates that expression of oncogenic *BRAF*^{V600E} gives rise to polyploidy in more than 90% of all zebrafish melanocytes expressing oncogenic BRAF. These data indicate that while oncogenic *BRAF*^{V600E} has the capacity to promote polyploidy in certain contexts, the species-specific factors that promote/prevent the accumulation of polyploid cells *in vivo* clearly need to be better elucidated. We include this discussion point in the revised manuscript.

“What happens to the ploidy of mouse or human melanocytes expressing BRAFV600E? Can YAP-5SA rescue the polyploidy phenotype? If YAP can rescue the decreased proliferation phenotype of BRAF expressing melanocytes, what is the ploidy of the cells that respond to YAP? Can BRAF polyploid cells proliferate in response to YAP-5SA?”

These are excellent questions. As mentioned above, it does not appear that expression of oncogenic *BRAF* dramatically increases the number of polyploid melanocytes in mouse nevi, as most of the nuclei in nevi appear mononucleated and diploid by confocal microscopy. This could be because mitosis following expression of *BRAF*^{V600E} is more efficient *in vivo* and/or that polyploid cells that are generated *in vivo* are quickly eliminated by cell death or the immune system (Senovilla et al, Science, 2012).

We have previously demonstrated that YAP-5SA expression can restore proliferation to polyploid cells both *in vitro* and *in vivo* (Ganem et al., Cell, 2014). As you point out, this raises the possibility that melanomas generated by *Lats1/2* knockout may tend to have a higher overall ploidy than tumors with an intact Hippo pathway. To address this question, we performed low-pass whole-genome DNA sequencing of mouse melanomas generated from *Braf*^{V600E}/*Lats1/2*^{-/-}, *Lats1/2*^{-/-}, and *Braf*^{V600E}/*PTEN*^{-/-} mice. We hypothesized that tumors from the *LATS1/2*^{-/-} background would be of higher ploidy. However, these data revealed that all tumors were near-diploid. We speculate that even though *LATS1/2* loss may enable the proliferation of polyploid cells, these ultimately get outcompeted by the more fit diploids. We now include this data in Supplementary Figure 8 and in our discussion.

We write: *“Our live-cell imaging also revealed that induction of oncogenic BRAF promotes chromosome alignment defects, prolonged mitosis, and mitotic or cytokinetic failures that lead to whole-genome doubling (WGD). Cells that have experienced a WGD (WGD⁺ cells) are genomically unstable and tumorigenic, and their contribution to human cancer is significant (75-77). Oncogenic BRAF may therefore facilitate tumorigenesis not only by activating MAPK signaling, but also by increasing the baseline level of oncogenic WGD⁺ cells. However, while ~30-40% of all human melanomas, and the vast majority of human metastatic melanomas, demonstrate evidence of WGD (78), our mouse melanoma models failed to exhibit this genomic feature (Figure S8). This is likely because the mouse models are already potently tumorigenic without the need for the additional oncogenic effects imparted by a WGD; nevertheless, the lack of any appreciable WGD in murine melanomas illustrates that these models have limitations in recapitulating the human disease (45, 76-78).*

“Fig3: The effects of BRAFV600E overexpression on stress fiber formation is not obvious as shown by the similar phalloidin staining (Fig3D). Also, the effects of ROCKi and RAC1i are indistinguishable in control or BRAFV600E expressing cells. I am convinced that there really is an effect, very underwhelming.”

We agree that the effects on the actin cytoskeleton are hard to appreciate by just looking at a limited number of stained cells. For that reason, we developed an unbiased method to quantitate the effects on stress fibers to make our data more rigorous. Nevertheless, to better show the effect, we now include higher resolution photos of representative cells along with arrows indicating the cytoskeletal changes. In addition, we also include phase-contrast images of the cells, as the cytoskeletal changes induced by BRAF expression are more conspicuous when looking at total cell morphology. These changes are included in Supplementary Figures 3 and 5.

We also agree that the ROCKi and RAC1i experiments were not robust experiments and have removed them from the study. In their place, we now present new data demonstrating that activation of MAPK leads to significant reductions in RhoA activity in melanocytes. We have included these significantly stronger data in Figure 4.

“It is difficult to appreciate tumor load in the absence of pictures showing whole tissues and tumors. The authors should provide pictures of Lats KO - B BRAFV600E mice (depilated/non-depilated) with tumors.”

We have now included images of the mouse melanomas in Figure 5 and Supplementary Figure 6.

“Why was zebrafish used to induce tumors with YAP-5SA in melanocytes when mice would be preferable?”

We agree that using a mouse model that expresses constitutive YAP would be preferable to zebrafish. However, we believed that the significant time/cost needed to perform that experiment in mice was not warranted based on the convincing zebrafish study that we were able to accomplish relatively quickly/affordably with our collaborators. We fully expect that expression of constitutively active YAP would similarly promote melanocyte transformation and tumorigenesis in mice.

“Questions remain as to how Hippo pathway activation so strongly promotes melanoma development” should rather be “Hippo pathway inactivation”?

Good catch! We have made the revision.

REVIEWERS' COMMENTS

Reviewer #1 (Remarks to the Author):

In their revisions to "Inactivation of the Hippo Tumor Suppressor Pathway Promotes Melanoma", Vittoria et al have extensively revised the manuscript and appear to have addressed each reviewer critique/comment either with additional experiments or modification of the existing text or figures. Overall, the study is of interest and significance to the field of melanoma biology and YAP/TAZ signaling, and the conclusions support the conclusions shared.

Reviewer #3 (Remarks to the Author):

We are pleased to see that the manuscript by Vittoria et al. is much improved since its original submission. The authors have shown improvements in both the organization of the data and the conceptual mechanism of how oncogene activation regulates Hippo signaling in tumor initiation. Overall, the revised manuscript presents a noteworthy and well-supported finding about melanoma initiation and should be accepted for publication after the few remaining minor comments below are addressed.

Figure 3d: What are these values normalized to? The legend does not say. Also, are there more time points that can be included from this experiment? It would be nice to see a more longitudinal curve of the data. The point is made as is, but it would be more convincing to have a richer dataset of cell growth over time.

Figure S3b: This is a nice new panel. How were the EdU-incorporated cells analyzed? Flow cytometry or fluorescence microscopy? The legend and methods do not say.

Figure S5b: A timeline schematic or clarification for when dox was added relative to ERKi would be helpful. This clarification may help with the discrepancy seen in p-YAP at the 8 h time point, which shows a 1.5-fold increase in Main Fig. 4d vs. a 5-fold increase in S5b. It is also surprising that the early time points 2-8 h experience minimal effects of the ERKi addition; can the authors reconcile why this is? Also why is the 24 h DMSO without dox 3-fold higher than the 2 h DMSO without dox?

Reviewer #4 (Remarks to the Author):

My questions have been addressed.

Reviewer #1: *In their revisions to “Inactivation of the Hippo Tumor Suppressor Pathway Promotes Melanoma”, Vittoria et al have extensively revised the manuscript and appear to have addressed each reviewer critique/comment either with additional experiments or modification of the existing text or figures. Overall, the study is of interest and significance to the field of melanoma biology and YAP/TAZ signaling, and the conclusions support the conclusions shared.*

We thank the reviewer for their previous comments/suggestions.

Reviewer #4: *My questions have been addressed.*

We thank the reviewer for their previous comments/suggestions.

Reviewer #3: *We are pleased to see that the manuscript by Vittoria et al. is much improved since its original submission. The authors have shown improvements in both the organization of the data and the conceptual mechanism of how oncogene activation regulates Hippo signaling in tumor initiation. Overall, the revised manuscript presents a noteworthy and well-supported finding about melanoma initiation and should be accepted for publication after the few remaining minor comments below are addressed.*

We thank the reviewer for their comments/suggestions.

Figure 3d: What are these values normalized to? The legend does not say. Also, are there more time points that can be included from this experiment? It would be nice to see a more longitudinal curve of the data. The point is made as is, but it would be more convincing to have a richer dataset of cell growth over time.

Figure 3d is a population doubling assay and represents the number of population doublings over a 4-day period relative to the initial plating density. Based on our protocol, these cells were plated at an initial density of 1×10^5 cells. To make this more clear, we have modified the figure legend: “Number of population doublings of indicated dox-inducible Mel-ST clone...”. Since counting cells requires trypsinizing and replating cells (which can induce error), we felt it was best to terminate the experiment after only counting to day 4, especially since we observed a statistically significant effect on cell growth at that time.

Figure S3b: This is a nice new panel. How were the EdU-incorporated cells analyzed? Flow cytometry or fluorescence microscopy? The legend and methods do not say.

We’ve expanded the EdU Assay subsection of the methods to include that the EdU-incorporated cells were analyzed via fluorescence microscopy.

Figure S5b: A timeline schematic or clarification for when dox was added relative to ERKi would be helpful. This clarification may help with the discrepancy seen in p-YAP at the 8 h time point, which shows a 1.5-fold increase in Main Fig. 4d vs. a 5-fold increase in S5b.

We agree. We have now clarified within the legends that initially no dox or dox was added for 24 hours and then subsequently treated with either DMSO or ERKi for 2 to 24 h.

It is also surprising that the early time points 2-8 h experience minimal effects of the ERKi addition; can the authors reconcile why this is?

In our main figures we demonstrate that activation of the Hippo pathway requires sustained MAPK activation for 16-24 h, and our data demonstrate it also takes time for the Hippo pathway to become

inactivated after disruption of MAPK signaling. This is what we believe is occurring in Figure S5b, where it takes time (8-16 h) after addition of the ERK inhibitor before significant rescue is appreciated. This is further evidence that the effect of BRAF activation on Hippo pathway activation is indirect.

Also why is the 24 h DMSO without dox 3-fold higher than the 2 h DMSO without dox?

Despite our best efforts to control for temporal differences in proliferation, we believe the 24 h DMSO without dox condition has elevated Hippo activation due to higher cell density secondary to increased time for proliferation.